# Modelling Chlamydia and HPV co-infection in patient-derived ectocervix organoids reveals distinct cellular reprogramming

Stefanie Koster [1,7], Rajendra Kumar Gurumurthy [1,7], Naveen Kumar [2], Pon Ganish Prakash[2], Jayabhuvaneshwari Dhanraj[2], Sofia Bayer[1], Hilmar Berger [1], Shilpa Mary Kurian[2], Marina Drabkina[1], Hans-Joachim Mollenkopf [1], Christian Goosmann[1], Volker Brinkmann [1], Zachary Nagel [3], Mandy Mangler[4,5], Thomas F. Meyer [1,6] & Cindrilla Chumduri [1,2✉]

Coinfections with pathogenic microbes continually confront cervical mucosa, yet their implications in pathogenesis remain unclear. Lack of in-vitro models recapitulating cervical epithelium has been a bottleneck to study coinfections. Using patient-derived ectocervical organoids, we systematically modeled individual and coinfection dynamics of Human papillomavirus (HPV)16 E6E7 and Chlamydia, associated with carcinogenesis. The ectocervical stem cells were genetically manipulated to introduce E6E7 oncogenes to mimic HPV16 integration. Organoids from these stem cells develop the characteristics of precancerous lesions while retaining the self-renewal capacity and organize into mature stratified epithelium similar to healthy organoids. HPV16 E6E7 interferes with Chlamydia development and induces persistence. Unique transcriptional and post-translational responses induced by Chlamydia and HPV lead to distinct reprogramming of host cell processes. Strikingly, Chlamydia impedes HPV-induced mechanisms that maintain cellular and genome integrity, including mismatch repair in the stem cells. Together, our study employing organoids demonstrates the hazard of multiple infections and the unique cellular microenvironment they create, potentially contributing to neoplastic progression.

[1] Department of Molecular Biology, Max Planck Institute for Infection Biology, Berlin, Germany. [2] Chair of Microbiology, University of Würzburg, Würzburg, Germany. [3] Harvard T.H. Chan School of Public Health, Boston, MA, USA. [4] Klinik für Gynäkologie und Geburtsmedizin, Vivantes Auguste-Viktoria-Klinikum, Berlin, Germany. [5] Department of Gynecology, Charité University Medicine, Berlin, Germany. [6] Laboratory of Infection Oncology, Institute of Clinical Molecular Biology (IKMB), Christian Albrechts University of Kiel, Kiel, Germany. [7] These authors contributed equally: Stefanie Koster, Rajendra Kumar Gurumurthy. ✉email: cindrilla.chumduri@uni-wuerzburg.de

Cervical mucosa being a protective barrier against invading pathogens into the upper female reproductive tract is often challenged by the invading pathogens and dysbiosis[1,2]. Coinfections are increasingly being discovered in clinical practice, yet their implication in disease development is unclear. The lack of in vitro systems that recapitulate the human cervical epithelium has been a bottleneck to decipher the individual and coinfection processes systematically and elucidate their role and impact in driving pathogenesis, including cervicitis and carcinogenesis. The recently established human and mouse-derived primary cervical epithelial three-dimensional (3D) organoids mimic the in vivo native tissue architecture and can be expanded and maintained long term[2–4]. Further, the ectocervical growth conditions enrich ectocervical stem cells in the 2D cultures. These 2D cultures could be an excellent source for stem cell-specific studies and certain applications like mass spectrometry that require large quantities of cells. These stem cells can be expanded in a short term, followed by organoid cultures in a cost-effective way[3,4]. Thus, these culture systems offer a unique possibility to elucidate the dynamics and the impact of different infections and coinfections in pathogenesis. The cervix comprises ectocervix lined by stratified squamous epithelium that projects into the vagina, and the endocervix lined by columnar epithelium that forms a continuum with the uterus. The ecto- and endocervix merge at the squamocolumnar transition zone, and these sites are highly predisposed to preneoplastic squamous metaplasia, a precursor of cervical cancers[1,4]. Most cervical cancers, the fourth most common cancers of women worldwide, originate from the squamous stratified epithelium[5].

HPV and bacterial pathogen Chlamydia trachomatis (C. trachomatis) are among the highly prevalent sexually transmitted infections. HPV has long been established as the etiological agent of cervical carcinogenesis[6,7]. Although HPV infections are encountered by over 80% of women during their lifetime, less than 2% develop cervical cancer[8,9]. Cofactors like immune status, hormones, and coinfections are emerging as a causal link in cervical cancer development[10–12]. Coinfections with C. trachomatis are seen at an increased incidence in patients with invasive cervical and ovarian cancers[10,13], yet the coinfection dynamics and the underlying mechanisms are entirely unknown. Unlike tumor viruses whose DNA can be found within the tumors, bacteria linked to cancers rarely leave any traceable elements. The integration of HPV E6E7 oncogenes into the host genome is often observed in high-grade lesions and is considered as a major trigger and point of no return during cervical cancer development[14]. One link to associate bacteria with the onset of cancers is identifying the cellular and mutational processes that contribute to cell transformation.

This study addressed two key aspects; first, we introduced a near-physiological patient-derived-ectocervical organoid model to study the interaction of stratified epithelial tissue barrier and individual and coinfection dynamics of HPV16 E6E7 and Chlamydia. Second, we systematically unraveled their impact on the host cellular and molecular processes. Human ectocervical stem cells derived from HPV-negative healthy donors were genetically manipulated to integrate HPV16 E6E7 DNA into the genome. These HPV negative and HPV E6E7 expressing stem cells and organoids were infected with C. trachomatis and characterized the infection process in a single and coinfection scenario. HPV E6E7 integration induces changes observed in cervical intraepithelial neoplasias (CIN) and promotes aberrant Chlamydia development. The global transcriptomic analysis revealed HPV E6E7 and C. trachomatis-induced unique host cellular reprogramming. Several genes were discovered to be similarly up or down-regulated by both pathogens involving specific immune responses. Strikingly, a significant subset of all regulated genes

controlled by an E2F transcription factor and associated with DNA mismatch repair (MMR) was discovered to be oppositely regulated by HPV E6E7 and C. trachomatis at the transcriptional and post-translational level. Together, this study goes beyond the state-of-the-art and shows the hazards posed by multiple or sequential infections and the molecular mechanisms contributing to carcinogenesis. Our cervical organoids prove to be excellent in vitro models for studying complex interactions between epithelial tissue and pathogens and analyzing the molecular sequels of initial events of pathogenesis.

## Results

**Human ectocervical organoids to mimic HPV persistence**. To overcome the current lack of a suitable epithelial primary cell model that recapitulates the human ectocervical stratified epithelium and model disease development, we recently established an adult stem cell-derived-ectocervical organoids[3,4]. The ectocervical stem cells isolated from healthy donors were embedded in Matrigel to culture 3D organoids. Alternatively, they were first cultured in collagen-coated cell culture flask with a defined cocktail of growth factors to enrich stem cells and were subsequently cultured on irradiated mouse fibroblasts (J2-3T3) to maintain the stem cells in long-term cultures (Fig. 1a). These stem cells were used for experiments or to generate the mature ectocervical 3D organoids comprising stem cells and differentiated cells.

To systematically evaluate the impact of HPV16 E6E7 integration and subsequent coinfection with C. trachomatis on the host regulation, it is essential to have epithelial organoids from individuals negative for high-risk HPV (hrHPV). Therefore, we established ectocervical organoids from healthy donors and tested for E6E7 genes from various HPV types. The ectocervical organoids from three donors were found to be negative for E6E7 oncogenes of all tested HPV (hrHPVs 16, 18, 31, 33, 35, 45, 51, 52, and 58 as well as noncarcinogenic HPVs 6, 11, 40, 42, 43, 44, and 54), and shown are the data for HPV 16 and 18 (Supplementary Fig. 1a), which account for over 70% of cervical cancers[15]. The healthy human ectocervical (hCEcto) tissue-derived lineage-specific cytokeratin 5 positive (KRT5+) epithelial stem cells can self-organize into 3D organoids resembling the stratified squamous epithelium of the parent tissue. These organoids comprise the basal progenitor, parabasal, intermediate, and terminally differentiated superficial layers. They express epithelial marker E-cadherin (CDH1) and are KRT5+ but negative for cytokeratin 8 (KRT8), a columnar epithelium marker of the endocervix. The outer layer of the organoid comprises p63+ and Ki67+ basal progenitors. They have the proliferative and regenerative capacity to produce differentiated epithelium by the continuous movement of cells from the basal to luminal layers[3,4] (Fig. 1b–d, Supplementary Fig. 1b).

Next, the organoids from these healthy donors negative for E6E7 from hrHPV strains were used to study the impact of HPV integration on the healthy cervical tissue. Towards this end, the hCEcto stem cells were transduced with lentiviruses carrying HPV16 E6E7 oncogenes to genetically modify these cells to integrate E6E7 into the host cell genome. As expected, the non-transduced hCEcto stem cells from donors 1–3 were negative for both HPV16 and HPV18, while the hCEcto cells transduced with lentiviruses were positive for HPV16 E6E7 in all three donors (Fig. 1e, Supplementary Fig. 1d, e). The hCEcto cells expressing HPV16 E6E7 (hCEcto E6E7) preserve their stemness and retain their capacity to grow into mature organoids under the same culture conditions as hCEcto epithelial stem cells. The hCEcto E6E7 cells could be expanded and long-term propagated both as 2D stem cell cultures on the feeder system and as fully matured

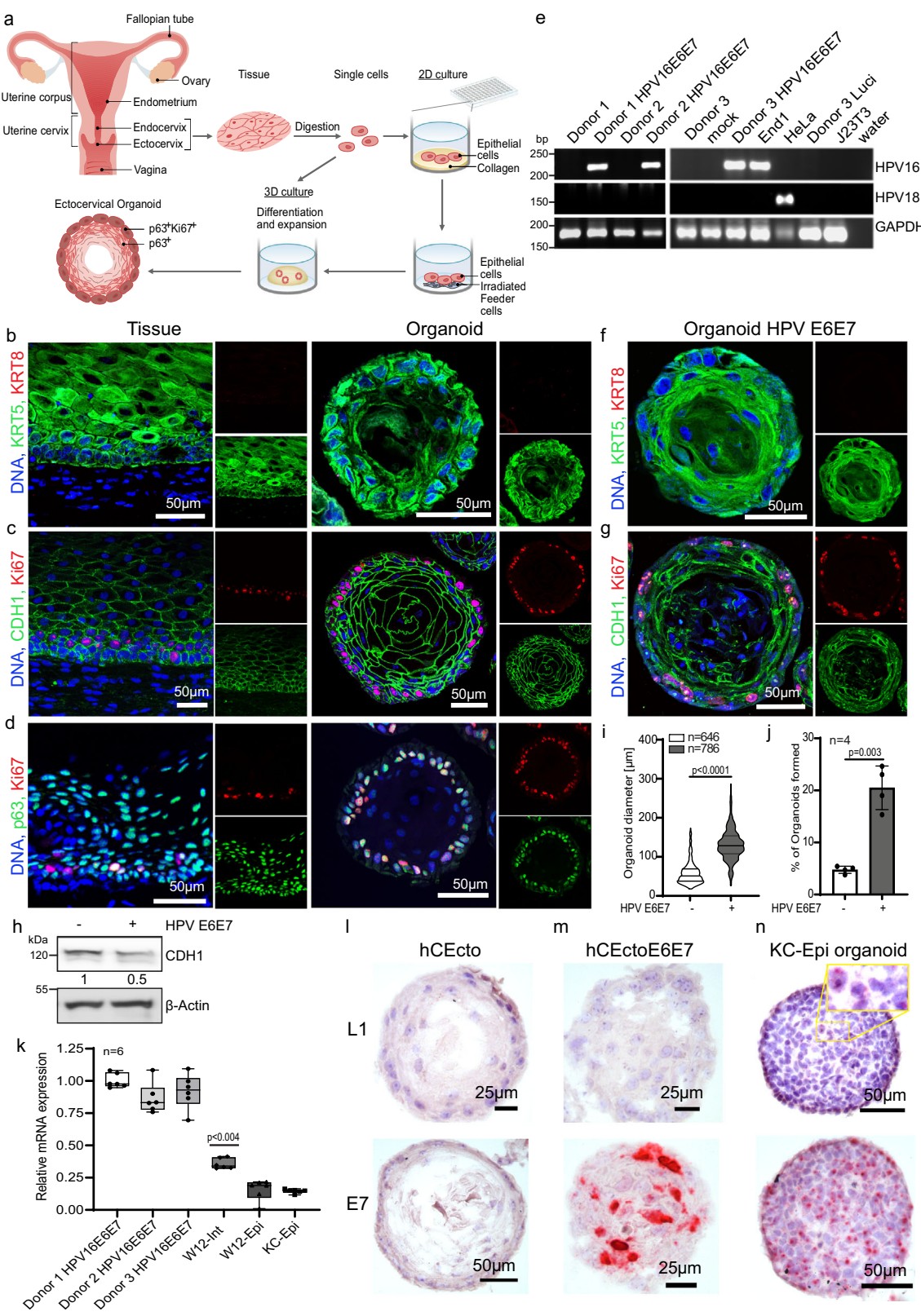

organoids. The two-week-old hCEcto E6E7 mature organoids consist of an outer layer of basal p63+ progenitors that are proliferative (Ki67+) and remain KRT5+/KRT8− (Fig. 1f, g, Supplementary Fig. 1c) similar to the normal ectocervical epithelium. However, a significant reduction and alteration in the distribution of adherens junction marker E-cadherin (CDH1), responsible for maintaining cell polarity and differentiation, was

observed in E6E7 positive organoids (Fig. 1c, g, h). They also show nuclear atypia with varied nuclear sizes (Fig. 1f, g, Supplementary Fig. 1c). hCEcto E6E7 organoids were found to proliferate faster, producing several layers of differentiated cells. The parabasal cells in the HPV E6E7 expressing organoids show expression of loricrin, a phenotype distinct from normal ectocervical epithelial organoids (Supplementary Fig. 1c). The

**Fig. 1 Genetic manipulation of HPV negative ectocervical organoids to integrate HPV16 E6E7 oncogenes. a** Schematic representation of human ectocervical organoid and 2D stem cell culture generation from biopsies. **b–d** Confocal images of human ectocervical tissue and organoids immunolabeled for KRT5, KRT8 (**b**), Ki67, E-cadherin (CDH1) (**c**); Ki67, p63 (**d**); Nuclei in blue. **e** PCR to detect HPV16, HPV18 E6E7, and GAPDH in samples from three independent experiments. Donor 1–3, untreated ectocervical stem cells; mock, Polybrene treated cells; Donor 1-3 HPV E6E7, Donor 1-3 transduced with HPV16 E6E7 lentivirus; End1, End1 cells; HeLa, HeLa cells; Donor 3 Luci, cells transduced with Luciferase lentivirus; J23T3, feeder cells; water, water control. **f, g** Confocal images of ectocervical organoids expressing HPV16 E6E7 immunolabeled for KRT5, KRT8 (**f**); Ki67, CDH1 (**g**); Nuclei are shown in blue. Images in (**b–d**) and (**f, g**) represent organoid cultures from three biological experiments. **h** Lysates of hCEcto and hCEcto E6E7 organoids were subjected to immunoblot analysis for CDH1 and β-actin as a loading control. Data represent three biological replicates. Densitometry values for CDH1 normalized to the β-actin, values represent the relative FC in hCEcto E6E7 compared to hCEcto. **i, j** Proliferation and stemness were measured by quantification of the diameter of hCEcto and hCEcto E6E7 organoids (**i**) and organoid formation (**j**). Data are presented as violin plot min–max, lines are 25 and 75th percentile, bold line is median (**i**); data are presented as mean values ± SD of four biological replicates (**j**). **k** Relative mRNA expression of HPV16 E6E7 analyzed by qRT-PCR from RNA extracts of three Donors (1–3) HPV E6E7, W12-Int or W12-Epi, and KC-Epi cells. Whiskers are min–max, box is 25–75th percentile, lines are median values normalized to the geometric mean of Donor 1–3 HPV E6E7. **l–n** smRNA-ISH images of organoids derived from hCEcto (**l**), hCEcto E6E7 (**m**), and KC-Epi (**n**) probed with HPV16 L1 and E7; Nuclei are shown in blue. Images are representatives of three replicates. Statistical significance was calculated using (**i, j**) two-sided *t* test, (**k**) One-way ANOVA, *P*-values are indicated. Source data are provided as a Source Data file.

faster proliferation of hCEcto E6E7 than hCEcto organoids is reflected in the significantly larger diameter of the mature organoids and the higher proportion of organoid formation ability (Fig. 1i, j). Together, integrating HPV16 E6E7 into ectocervical organoids induces a phenotype with features reminiscent of CINs.

Further, we evaluated the relative expression levels of HPV E6E7 in the transduced hCEcto cells compared to W12 cell lines derived from a patient with a low-grade cervical lesion that contained HPV in an episomal state (W12-epi) or with HPV E6E7 integration (W12-int) and primary keratinocytes containing episomal HPV16 genome (KC-epi) by qRT-PCR analysis. We found that the E6E7 expression levels are higher in hCEcto E6E7 and W12-int cells than in W12-epi and KC-epi cells. The E6E7 expression in hCEcto cells is relatively higher than in the W12-int cells (Fig. 1k). Strong expression of the E7 gene in hCEcto E6E7 cells was confirmed by single-molecule RNA in situ hybridization (smRNA-ISH) analysis. However, no signal for late gene L1, essential for virion assembly, could be detected as expected (Fig. 1l, m). In contrast, organoids generated from KC-epi cells expressed E7 as well as L1 genes in the luminal cells indicating that the organoid model supports a productive HPV life cycle (Fig. 1n). Thus, these organoid culture conditions open an avenue to study HPV and coinfection biology.

**Modelling *C. trachomatis* coinfection in HPV16 E6E7 expressing organoids.** Next, we sought to establish the ectocervical organoids as *C. trachomatis* infection model and investigate if these organoids support *C. trachomatis* infection and developmental life cycle. For this purpose, we first established *C. trachomatis* infection protocols in the ectocervical organoids. *C. trachomatis* has a biphasic life cycle. The non-replicative elementary bodies (EBs) infect the cells and transform into non-infectious replicative reticulate bodies (RBs). At the time of exit from the host cells, these RBs redifferentiate to EBs to initiate new infections. Efficient infection of both hCEcto and hCEcto E6E7 organoids was achieved by incubating intact small organoids (5 days old) for 2 h with *C. trachomatis* EBs and reseeding them back into Matrigel to allow these organoids to continue their growth. To follow the progression of *C. trachomatis* infection in organoids, we performed live-cell imaging and used recombinant *C. trachomatis* L2 strain expressing enhanced green fluorescent protein (GFP-*Ctr L2*) for detection in the organoids (Fig. 2a–d). First, GFP signals from the developing inclusion were detected in basal and parabasal cells of hCEcto and hCEcto E6E7 from 16 h post-infection (hpi) and 20hpi, respectively. The progression of *C. trachomatis* infection from the basal stem cell layer was observed to be bidirectional. It progressed from the basal cells to daughter

cells that self-renew by symmetric division and remain in the basal layer as stem cells, and the ones that form the differentiated epithelial cells via asymmetric division of the stem cells and move towards the lumen (Fig. 2a, b). Thus, infected stem cells can be a potential reservoir for *C. trachomatis* to propagate infection in the host. At 5 days post-infection (dpi), strong GFP signals were detected across all differentiated cell layers, and inclusion size increased significantly (Fig. 2c, d; Supplementary Fig. 2a). Further, we performed an infectivity assay to evaluate if *C. trachomatis* can complete its developmental cycle by redifferentiation from RBs to EBs within the organoids. The EBs obtained from the lysate of infected organoids at 1dpi or 5dpi were used to infect a monolayer of HeLa cells. Quantification of *C. trachomatis* inclusion forming units revealed a ten-fold increase in infectivity from lysates-derived from organoids infected for 5 days compared to 1 day (Fig. 2e, f), reflecting the increase in the infection load with time. A significant increase in inclusion size and infectious progenies shows *C. trachomatis* replicative activity and completion of its developmental life cycle in ectocervical organoids.

Next, we investigated the impact of HPV16 E6E7 expression on *C. trachomatis* development. Although both hCEcto and hCEcto E6E7 cells could be infected with *C. trachomatis* similarly in organoids or 2D stem cells as shown by inclusion formation (Fig. 2g, Supplementary Fig. 2c), HPV E6E7 suppressed the intracellular development of *C. trachomatis* shown by smaller inclusion size (Fig. 2h, Supplementary Fig. 2b). Further, *C. trachomatis* from the HPV E6E7 expressing organoids revealed the loss of reinfection ability compared to hCEcto organoids (Fig. 2i; Supplementary Fig. 2c), showing that the expression of HPV E6E7 interferes with the normal development of *C. trachomatis*.

To evaluate the influence of HPV E6E7 on different development stages of *C. trachomatis*, we performed transmission electron microscopy of Chlamydia infected hCEcto and hCEcto E6E7 cells. The ultrastructural analysis revealed that the inclusions contain a mixture of particles from different *C. trachomatis* developmental stages. In hCEcto cells, reproductive RBs mostly located at the periphery, condensed infective EBs, and the intermediate stage (IB) were detected (Fig. 2j). These observations are in line with earlier reports[16]. In hCEcto E6E7, besides the EB, RB, and IBs, we found an increased number of aberrant Chlamydial developmental forms (AB). As described previously[17], these ABs are membrane-lined, roundish, pleomorphic or irregularly shaped, and less electron-dense or hollow because of loss of cytoplasmic content and appear like ghost bacterial structures (Fig. 2k). Quantification of the various developmental forms revealed that the proportion of all *C.*

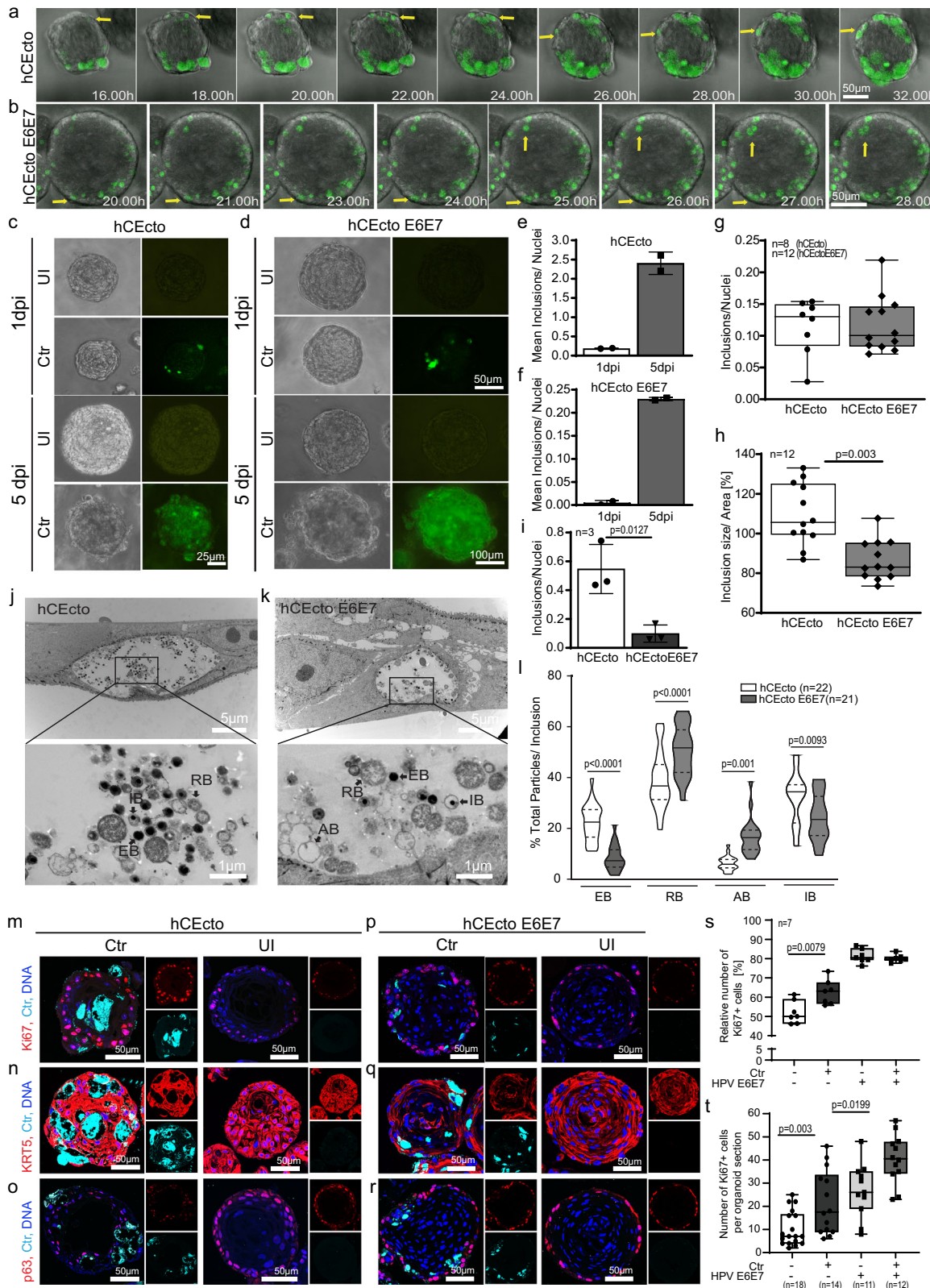

*trachomatis* stages is significantly different between E6E7 positive and negative cells. The number of infectious EBs is significantly reduced in E6E7 expressing primary cells, while the proportion of RBs and ABs is enhanced (Fig. 2l), corroborating with the reduced infectivity observed in these cells (Fig. 2i). These data reveal that HPV16 E6E7 increases AB formation and simultaneously inhibits RB to EB redifferentiation, resulting in a

slowdown of *C. trachomatis* development life cycle initiating persistence.

To assess the effects of *C. trachomatis* on host cell proliferation and epithelial architecture, we performed immunofluorescent staining for Ki67, KRT5, p63, and *C. trachomatis* major outer membrane protein (MOMP) in hCEcto and hCEcto E6E7 organoids infected for 5 days with *C. trachomatis*. MOMP

**Fig. 2 Modelling HPV16 E6E7 integration and *C. trachomatis* coinfection in ectocervical organoids. a, b** Time-lapse live imaging of hCEcto (**a**) and hCEcto E6E7 (**b**) organoids infected with GFP-expressing *C. trachomatis* (GFP-Ctr). Representative time-lapse microscopic images at indicated time points after Ctr infection. Arrows highlight the progression of Ctr infection from basal cells to daughter cells in the basal and differentiated layers. **c, d** Representative images of hCEcto (**c**) and hCEcto E6E7 (**d**) organoids at 1 and 5dpi with GFP-Ctr of three biological replicates. **e, f** Infectivity of lysates from Ctr infected hCEcto (**e**) and hCEcto E6E7 (**f**) organoids at 1 and 5dpi. Shown is the mean ± SD of two replicates representative of three biological replicates. **g, h** Primary infection of Ctr infected hCEcto and hCEcto E6E7 cells (**g**) and inclusions size (**h**). Whiskers are min–max, box is 25–75th percentile, lines are median values. n = biological replicates. **i** Infectivity of Ctr infected hCEcto and hCEcto E6E7 2D stem cells lysates at 48 hpi. Shown is the mean ± SD of three technical replicates. **j, k** Transmission electron micrographs of Ctr infected hCEcto (**j**) and hCEcto E6E7 (**k**) 2D stem cells at 24hpi. Inclusion contains different development forms of Ctr (RB, reticulate bodies; EB, elementary bodies; IB, intermediate stage; AB, aberrant bodies). Shown are representative images from three biological replicates. **l** Percentage of total EBs, RBs, IBs, and ABs in inclusions, representative of three biological replicates. Data are presented as violin plot min–max, lines are 25 and 75th percentile, bold line is median. Confocal images of hCEcto (**m–o**) and hCEcto E6E7 (**p–r**) organoids at 5dpi immunolabeled for Ki67 (**m, p**); KRT5 (**n, q**), p63 (**o, r**) and Ctr-MOMP, Nuclei in blue. Representative images of three biological replicates. **s, t** Quantification of the number of Ki67 positive cells from confocal images of uninfected and Ctr infected hCEcto and hCEcto E6E7 2D stem cells at 48hpi (**s**) and organoids at 5dpi (**t**). Whiskers are min–max, box is 25–75th percentile, lines are median values. (**g–i, l, s, t**) Statistical significance was calculated using two-sided *t* test, *P*-values are indicated. Source data are provided as a Source Data file.

staining revealed large inclusions across different epithelial layers and inside the lumen (Fig. 2m–r). Both hCEcto and hCEcto E6E7 remain KRT5 + (Fig. 2n, q). However, an increased number of Ki67+ cells were observed, indicating an enhanced proliferation in *C. trachomatis* infected stem cells and organoids compared to uninfected organoids (Fig. 2m, s, t). Similar changes were observed in hCEcto E6E7 organoids coinfected with *C. trachomatis* (Fig. 2p, s, t). These results show that *C. trachomatis* infection induces proliferative signals in single and coinfections of HPV E6E7 expressing ectocervical epithelium.

***C. trachomatis*, HPV E6E7, and coinfection elicit unique transcriptional responses in human ectocervical organoids.** Next, we performed the transcriptomic analysis to identify the impact of *C. trachomatis*, HPV E6E7, and coinfection on cellular signaling. Towards this, hCEcto and hCEcto E6E7 cells were infected with *C. trachomatis* for 48 h in 2D stem cell cultures or 5 days in the 3D organoids. All genes differentially up or down-regulated (log2FC > 2, and adj. *p*-value < 0.05) upon *C. trachomatis*, HPV E6E7, and coinfection compared to uninfected HPV negative healthy cells were scored. Overall, our analysis revealed a massive transcriptional response to HPV E6E7 expression and *C. trachomatis* infection, respectively (Fig. 3a, Supplementary Fig. 3a, Supplementary Data 1). In 2D stem cells, HPV E6E7 differentially regulated 3456 transcripts, while *C. trachomatis* infection led to differential expression of 7239 transcripts. Among these, 717 genes are similarly regulated by both pathogens, while 706 genes are oppositely regulated (Fig. 3b, Supplementary Data 2). In the 3D organoid model, we identified 3139 transcripts differentially regulated by HPV E6E7 expression, while *C. trachomatis* infection led to differential expression of 2742 transcripts; of these, 610 genes are similarly regulated, and 203 genes are oppositely regulated by HPV and *C. trachomatis* (Supplementary Fig. 3b, Supplementary Data 2). Further, coinfections of hCEcto E6E7 cells with *C. trachomatis* exhibited unique gene expression changes. Strikingly, of the genes upregulated by HPV E6E7, 28% reverted to expression levels similar to that of control or were strongly downregulated when coinfected with *C. trachomatis*.

Gene Set Enrichment Analysis (GSEA) revealed several transcription factors (TF) are oppositely regulated by *C. trachomatis* and HPV E6E7. Strikingly, E2F family members were the most significant oppositely regulated TF among all comparisons (Fig. 3c, Supplementary Fig. 3c, Supplementary Data 3). E2F target genes are significantly upregulated by HPV E6E7 but downregulated by *C. trachomatis* in hCEcto organoids (Supplementary Fig. 3d, e). In coinfections, *C. trachomatis* overrides the HPV E6E7 induced E2F target genes expression

(Supplementary Fig. 3f). It is known that HPV E7 unleashes the E2F TF from retinoblastoma (RB) inhibitory interaction to activate the transcription of downstream targets. However, to date, not much is known regarding Rb-E2F1 pathway modulation by *C. trachomatis*. Further, it is unclear what consequences this *C. trachomatis* mediated suppression of HPV E6E7 induced E2F targets will have on host cells and pathogens.

To investigate the regulation of the Rb-E2F1 pathway by both pathogens, we infected hCEcto and hCEcto E6E7 with *C. trachomatis* and performed qRT-PCR and immunoblot analysis (Fig. 3d–g). qRT-PCR analysis revealed that *E2F1* and tumor suppressor p53 (*TP53*) are significantly suppressed by *C. trachomatis* in the absence of HPV E6E7 expression, while coinfection results in a significant reduction of *TP53*, but not *E2F1* and *RB1* transcripts. In contrast to *C. trachomatis* infection, HPV E6E7 upregulated *TP53*, *E2F1*, and *RB1* gene expression (Fig. 3d–f). Immunoblot analysis revealed the reduction of total Rb, pRb (S807/811), E2F1, and p53 proteins upon *C. trachomatis* infection in hCEcto cells irrespective of HPV status (Fig. 3g). However, E6E7 expression increases E2F1 protein levels and induces near complete degradation of p53 while having no significant effect on total Rb, pRb (S807/811) protein levels (Fig. 3g) in agreement with previous studies[18,19]. Together, our data suggest HPV induces E2F1 activation, but Chlamydia suppresses it. HPV and Chlamydia lead to p53 protein degradation. However, p53 is upregulated by HPV at the transcriptional level while downregulated by Chlamydia in single and coinfections.

Next, we assessed the biological functions affected by the differentially expressed genes (DEG) by performing GSEA analysis between uninfected, *C. trachomatis* infected, E6E7 expressing, and coinfected hCEcto cells to identify overrepresented networks from Kyoto Encyclopedia of Genes and Genomes (KEGG) and Gene Ontology (GO) terms. The tumor necrosis factor (TNF) pathway was strongly upregulated by both HPV E6E7 and *C. trachomatis*. Further, specific pathways were uniquely regulated by each pathogen. The inflammatory response (e.g., Interleukin-17 and nuclear factor (NF)-kappa B signaling) and mitogen-activated protein kinase (MAPK) pathways are distinctly upregulated by *C. trachomatis*, while pathways like oxidative phosphorylation, RNA regulation, and RNA processing are downregulated. HPV E6E7 expression was shown to have a negative impact on differentiation processes, increased mitotic activities, RNA and DNA replication (Supplementary Fig. 3h,i, Supplementary Data 4). To gain further insights into the effect of HPV E6E7 and Chlamydia on the stemness and differentiation in the ectocervical epithelium, we compared the cervical stem cell and differentiation-specific transcriptional signatures from[4] to the

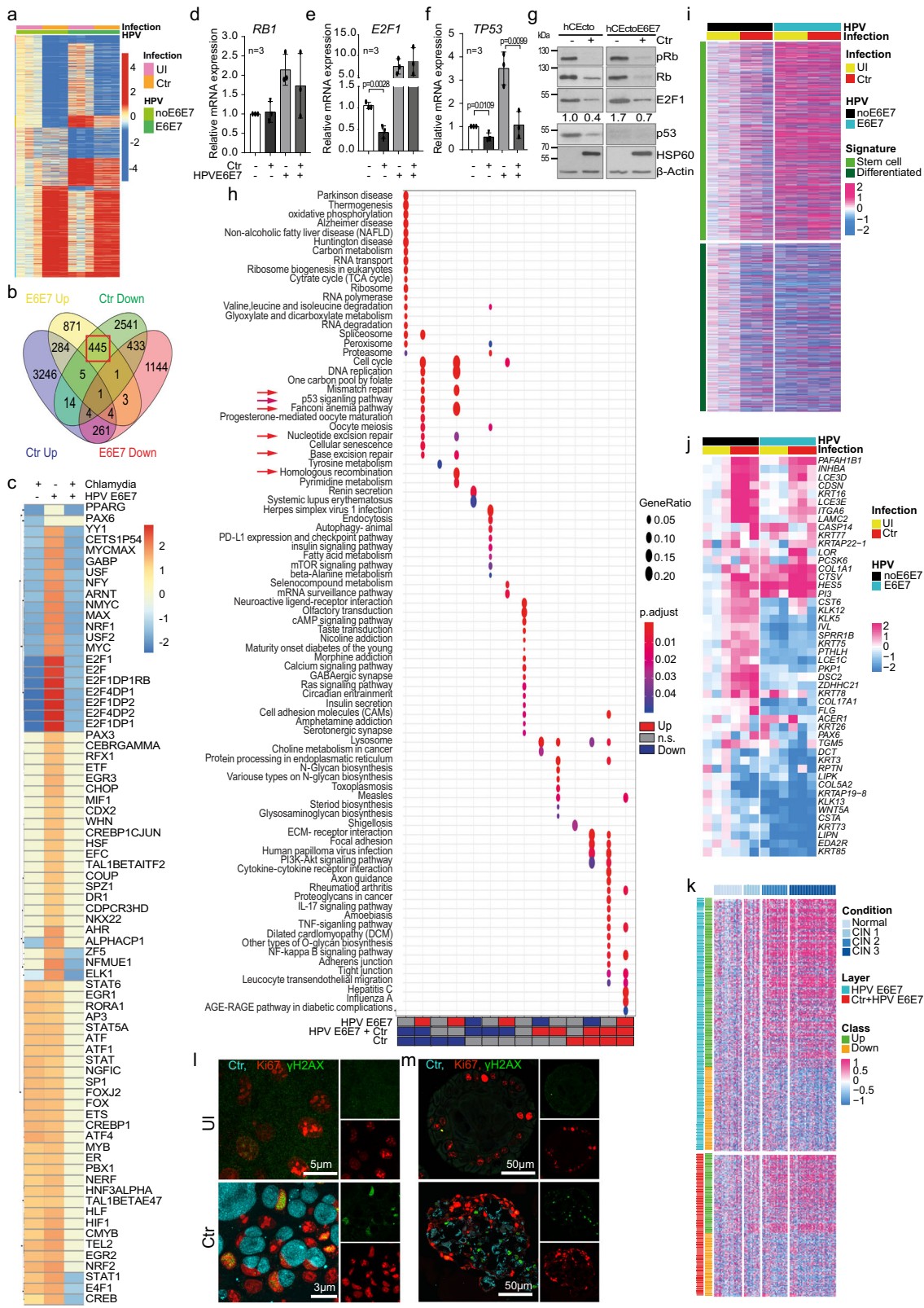

gene expression induced by HPV E6E7 and Chlamydia infections. We found that HPV E6E7 and Chlamydia infection both upregulate genes associated with stemness and down-regulate differentiation-associated genes (Fig. 3i). Further, we extracted the genes associated with skin development, epidermis development, keratinocyte differentiation, keratinization, epidermal cell differentiation, and cornification GO terms that were oppositely

regulated in HPV E6E7 and Chlamydia infections shown in Supplementary Fig. 3h. The transcription of a subset of these differentiation-associated genes is oppositely regulated, while others are synergistically regulated by HPV E6E7 and Chlamydia (Fig. 3j). Together the data shows that both HPV E6E7 and Chlamydia promote stemness and reduce differentiation. To gain a deeper molecular pathological understanding of HPV E6E7 and

**Fig. 3 Host transcriptional response induced by *C. trachomatis*, HPV16 E6E7 and coinfection. a** Heatmap of DEG in hCEcto and hCEcto E6E7 2D stem cells with or without Ctr infection; Columns represent biological replicates. **b** Venn diagram shows genes regulated in hCEcto 2D stem cells after Ctr infection or HPV E6E7 expression. Red box highlights genes upregulated by HPV E6E7 and downregulated by Ctr. **c** Heatmap showing GSEA enrichment (–log10(*P*-value)) of genes that share Cis-regulatory motifs for transcription factors. **d**–**g** hCEcto and hCEcto E6E7 2D stem cells were infected with Ctr for 48h. mRNA expression of *RB1* (d), *E2F1* (**e**), and *TP53* (**f**) analyzed by qRT-PCR. Data represent mean ± SD from three biological replicates. Statistical significance was calculated using two-sided *t* test, *P*-values are indicated. (**g**) Immunoblot analysis for indicated proteins; β-actin. Densitometry values for E2F1 normalized to β-actin; representing FC compared to hCEcto. Data represent three biological replicates. **h** Dot plot showing up-or down-regulated KEGG pathways among DEG from hCEcto and hCEcto E6E7 2D stem cells with or without Ctr infection. Red arrows highlight DNA repair pathways. Dot diameter refers to the gene ratio within a group. Fill color depicts the adjusted *P*-value. **i** Heatmap of genes upregulated in ectocervical stem and differentiated cells from ref. [4] correlated with expression profiles of hCEcto and hCEcto E6E7 organoid with or without Ctr infection. **j** Heatmap showing expression of genes associated with GO terms for cornification, epidermal cell differentiation, keratinization, skin and epidermis development in hCEcto and hCEcto E6E7 organoids with or without Ctr infection. **k** Heatmap of DEG from hCEcto E6E7 organoids with or without Ctr infection and their corresponding expression in normal and CIN1-3 tissues. The color bar depicts expression values after subtracting the mean of uninfected hCEcto samples (**i**, **j**) and normal cervical tissue samples (**k**) and dividing by the SD of each probe. **l**, **m** Confocal images of hCEcto 2D stem cells (**l**) and organoids (**m**) infected for 36 h and 5 d, respectively, with Ctr and immunolabelled for Ki67, γH2AX, and Chlamydia MOMP. **l**, **m** Data represent three biological replicates. Source data are provided as a Source Data file.

Chlamydia coinfections induced transcriptional alterations, we have performed a comparative analysis of HPV E6E7 and coinfection induced up or downregulated genes signatures to expression levels of these genes in healthy, and CIN1-3 tissues obtained from[20]. The analysis revealed that the HPV E6E7 and Chlamydia coinfection induce similar transcriptional profiles observed in CINs (Fig. 3k).

Intriguingly, pathways involved in DNA replication, DNA damage repair pathways including base excision repair (BER), mismatch repair (MMR), nucleotide excision repair (NER), Fanconi anemia (FA), homologous recombination (HR), and p53 signaling pathway are oppositely regulated by the two pathogens: HPV E6E7 expression activates these pathways, while *C. trachomatis* suppress them (Fig. 3h, Supplementary Fig. 3g, Supplementary Data 5). Further, we show that Chlamydia induces DNA double-strand breaks in stem cells and differentiated cells while promoting cellular proliferation, as shown by an increased number of Ki67 and phosphorylated histone H2AX (γH2AX) positive cells (Figs. 2s, t, 3l, m).

**C. trachomatis suppresses HPV-induced mismatch repair pathway in ectocervical stem cells.** Since the E2F family of transcription factors, vital regulators of several genes involved in cell proliferation and growth arrest, are downregulated by *C. trachomatis* and coinfection, we further analyzed the significance of their altered regulation on the Chlamydia infected cells. RB-like, E2F4, multi-vulval class B proteins and dimerization partner (DP1) form the DREAM complex that mediates gene repression promoting quiescence. Disruption in the DREAM complex-mediated regulation switches the balance from quiescence to proliferation by losing cell cycle checkpoint gene expression, which is frequently observed in cancers[21,22]. Our analysis revealed that many DREAM complex target genes are downregulated by Chlamydia, and in coinfection, that could prevent quiescence and promote cell proliferation, which is corroborated by increased Ki67+ cells upon Chlamydia infections (Figs. 4a, 2s, t, 3l, m, Supplementary Data 6). Notably, genes involved in DNA repair pathways are among the E2F1 target genes oppositely regulated by Chlamydia and HPV (Fig. 4a, b red boxes, Supplementary Data 7).

Mutational signatures associated with specific mutational processes have been found in various human cancers; two signatures attributed to defective DNA mismatch repair[23,24] are found in cervical cancer[25,26]. However, MMR, which plays a crucial role in preventing mutations because of misincorporation of bases that can arise during DNA replication and recombination[27], is upregulated by HPV E6E7 but downregulated

by *C. trachomatis* infection. *C. trachomatis* suppressed the HPV E6E7 induced MMR pathway (Figs. 4c, 3h, Supplementary Fig. 3g). We found that most of the genes involved in MMR, including MutL homolog 1 (*MLH1*), MutS homolog 2 (*MSH2*), and MutS homolog 6 (*MSH6)* that play a central role in mismatch detection and initiation of the MMR signaling process[28], are strongly downregulated by *C. trachomatis* infection in both hCEcto stem cells and organoids (Supplementary Fig. 4a). *C. trachomatis* infection of HPV E6E7 expressing cells leads to a suppression of HPV-induced MMR genes (Fig. 4d, e). Corroborating the global transcriptomic data, MSH6 and MLH1 protein levels were upregulated by HPV E6E7 but reduced by Chlamydia in single and HPV coinfections (Fig. 4f, Supplementary Fig. 4b, c). We confirm that MMR protein downregulation in Chlamydia infection is not an artifact resulting from Chlamydial protease-like activity factor (CPAF)-mediated post cell lysis degradation of these proteins as the control protein golgin-84 remains uncleaved (Supplementary Fig. 4d). Thus, Chlamydia suppresses MMR at both transcriptional and post-translational levels, and it can further suppress HPV-activated MMR.

Since organoids are complex epithelial structures composed of different cell populations and differentiation stages, we investigated the spatial distribution of the MMR protein MSH6 in normal and *C. trachomatis* infections. Confocal images of the organoid sections from uninfected or 5dpi revealed high expression of MSH6 in the nucleus of basal stem cells with higher intensities in hCEcto E6E7 organoids corroborating with the increased protein levels. Further, a complete loss of MSH6 was observed in the *C. trachomatis* infected hCEcto organoids (Fig. 4g). Interestingly, in hCEcto E6E7 organoids, *C. trachomatis* infected and neighboring cells show reduced or complete loss of MSH6 signals (arrows) (Fig. 4h). Next, we sought to analyze if the impaired MMR pathway culminates in altered MMR capacity within the *C. trachomatis* infected host cells. A functional assay employing a G:G mismatch containing reporter plasmid that expresses a nonfluorescent protein (mOrange) was used to investigate MMR efficiency[29]. Efficient repair of the in vitro generated G:G mismatch restores the wild type cytosine in the transcribed strand of the plasmid leading to mOrange fluorescent protein expression (Fig. 4i). To determine transfection efficiency, hCEcto and hCEcto E6E7 cells were simultaneously transfected with a G:G mismatch-containing reporter plasmid and an undamaged reporter plasmid (expressing mPlum) prior to *C. trachomatis* infection. Fluorescence intensity was measured 24hpi by flow cytometry in living cells. The data showed that *C. trachomatis* infection leads to impaired MMR repair efficiency in single and HPV E6E7 (co-) infections,

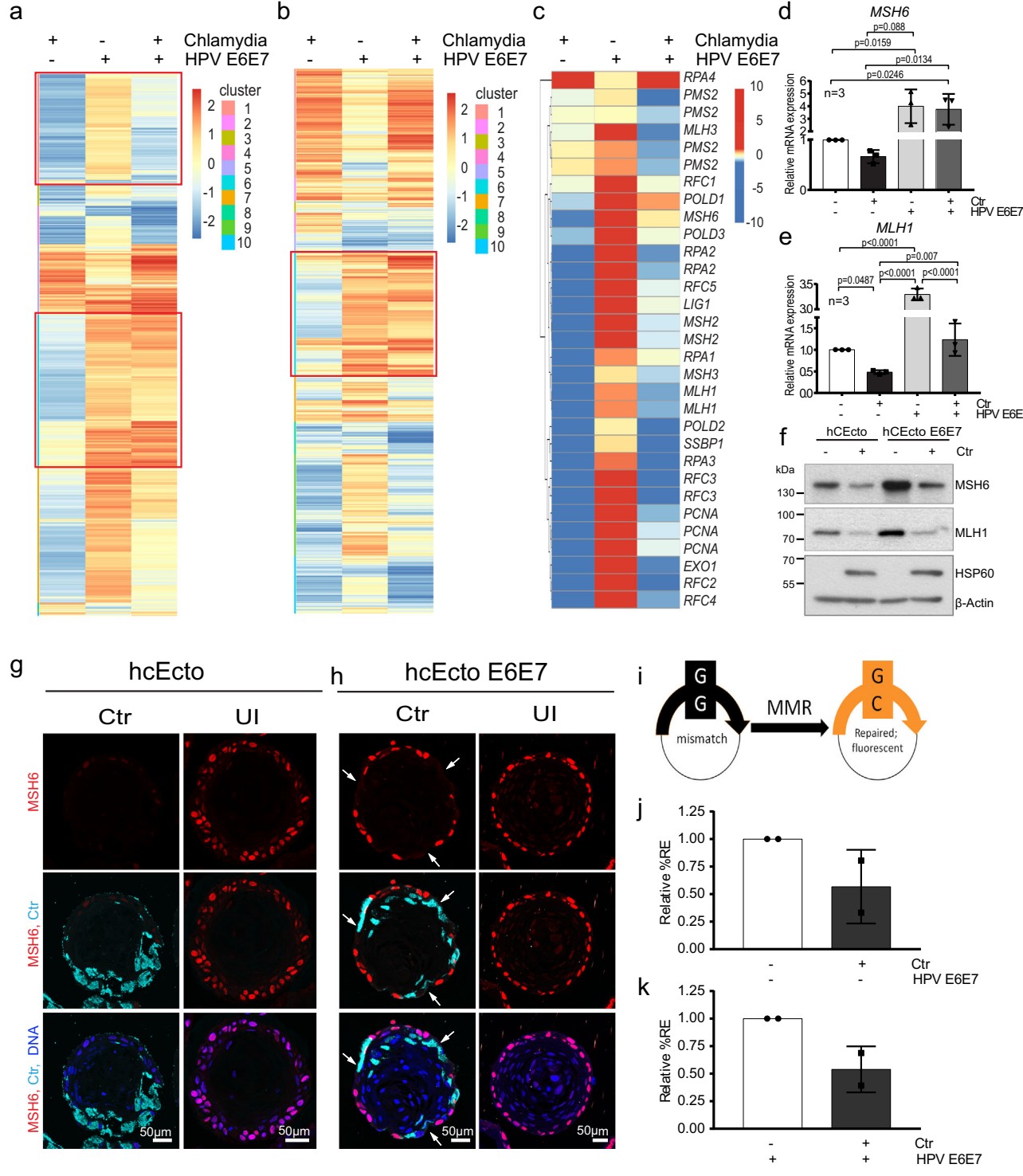

reflected by less reporter expression (%RE) in infected samples (Fig. 4j, k).

**E2F-p53 and proteasomal degradation diverges the transcriptional and post-translational regulation of MMR during coinfections.** Further, we sought to gain a mechanistic understanding of how Chlamydia and HPV regulate the MMR pathway. MMR genes are regulated by E2F transcription factors[30,31]. There is extensive crosstalk between the Rb–E2F and MDM2–p53 pathways, and remarkably, both are defective in most human tumors, which underscores the crucial role of these pathways in

regulating vital cellular decisions. Using well-established proteasome inhibitors MG-132 and Lactacystin to inhibit cellular and Chlamydia protease CPAF[32], we tested the expression levels of MMR transcripts. Next, we checked the effect of p53 stabilization on regulating MMR gene expression and protein levels using Nutlin-3a that inhibits the interaction between MDM2 and p53 to stabilize p53[33]. qRT-PCR analysis showed that Nutlin-3a and MG-132 treatment restored MMR gene expression after *C. trachomatis* infection (Fig. 5a, b). Lactacystin treatment increased *MLH1* expression moderately compared to the control, but not *MSH6*. However, treatment with all three inhibitors created an

**Fig. 4 Inverse regulation of mismatch repair pathway by *C. trachomatis* and HPV E6E7 in ectocervical stem cells and organoids. a** Heatmap showing all DREAM target genes arranged into ten clusters based on their expression pattern between comparisons among the DEG in hCEcto and hCEcto E6E7 2D stem cells with or without Ctr infection for 48 h. Red frames highlight the cluster containing mismatch repair (MMR) genes. Color bar depicts Z-scored expression values. **b** Heatmap showing E2F target genes arranged into ten clusters based on their expression pattern between comparisons among the DEG in hCEcto and hCEcto E6E7 2D cells with or without Ctr infection for 48 h. Red frames highlight the cluster containing MMR genes. Color bar depicts log2 FC. **c** Heatmap depicting the expression of MMR pathway genes. **d**, **e** hCEcto and hCEcto E6E7 2D stem cells were infected with Ctr for 48 h. Shown is the relative mRNA expression of *MSH6* (**d**) and *MLH1* (**e**) analyzed by qRT-PCR. Data represent mean ± SDs from three biological replicates normalized to uninfected controls. Statistical significance was calculated using one-way ANOVA, *P*-values are indicated. **f** hCEcto and hCEcto E6E7 2D stem cells were infected with Ctr for 48 h, and cell lysates were subjected to immunoblot analysis for indicated proteins and Chlamydia HSP60 and β-actin as a loading control. Data represent four biological replicates. **g**, **h** Shown are the hCEcto (**g**) and hCEcto E6E7 (**h**) organoids with or without Ctr infection for 5d and immunolabeled for MSH6, Ctr-MOMP, and Nuclei in blue. The arrow highlights the loss of MSH6 expression in hCEcto E6E7. Representative images of three biological replicates. **i–k** A reporter plasmid containing a G:G mismatch was used for fluorescence-based multiplex flow-cytometric host cell reactivation assay (FM-HCR) to analyze MMR efficiency by the restoration of reporter protein expression. **i** Comparison of relative reporter expression [% RE] between uninfected and Ctr infected hCEcto 2D stem cells (**i**) and hCEcto E6E7 (**j**) cells at 24hpi showing reduced mismatch repair efficiency after infection. Data represent mean ± SDs from two independent experiments normalized to uninfected controls. Source data are provided as a Source Data file.

unfavorable environment for *C. trachomatis* growth within the host cells, shown by a significant reduction in inclusion size compared to untreated control (Supplementary Fig 5a). Further, immunoblot analysis revealed that *C. trachomatis* induced p53 degradation is inhibited by Nutlin-3a and MG-132 treatment, while E2F1 degradation was reduced in all treatment conditions. Of note, the MMR proteins MLH1 and MSH6 were restored only by inhibiting proteasomal activity with MG-132 (Fig. 5c). The data show that Chlamydia engages p53-independent proteasomal degradation to reduce E2F1 and MMR proteins, while restoration of E2F1 and p53 proteins rescues transcriptional suppression of MMR genes (Fig. 5g).

To further investigate how *C. trachomatis* regulates the MMR pathway in HPV E6E7 coinfections, hCEcto E6E7 stem cells were treated with Nutlin-3a, MG-132, and Lactacystin with or without *C. trachomatis* infection for 48 h. Like hCEcto cells, all treatment conditions reduced *C. trachomatis* infectivity as shown by significantly smaller inclusions than in control (Supplementary Fig. 5b). In contrast to hCEcto cells, Nutlin-3a treatment of *C. trachomatis* infected hCEcto E6E7 cells shows further suppression of *MLH1* or *MSH6* gene expression (Fig. 5d, e). However, proteasome inhibition by MG-132 led to a marginal rescue of MLH1 or MSH6 gene and protein expression. Further, Nutlin-3a and MG-132 treatment rescued the E2F1 protein expression. Notably, inhibition of proteasomal degradation by MG-132 stabilized p53, while Nutlin-3a and Lactacystin treatment do not influence HPV-mediated degradation of p53 (Fig. 5f). Together, proteasomal degradation is a crucial mechanism that distinctly regulates E2F-mediated MMR gene transcription and reduces p53, E2F, and MMR protein in coinfections (Fig. 5g).

## Discussion
Insights into the coinfection of viruses and bacteria resulting in disease exacerbation are emerging[34,35]. Mucosal surfaces of the female reproductive tract (FRT) play a central role in protecting the reproductive tract from infections while maintaining tissue homeostasis to prepare for successful fertilization and childbirth. Since it is critical to maintain sterile conditions in the upper FRT comprising the uterus, fallopian tube, and ovaries, the lower FRT, particularly the cervix, acts as a gatekeeper between the uterus and the vagina bearing the burden of defending against ascending infections. Despite the enormous importance of the cervical epithelium in maintaining women's health and human reproduction, the lack of in vitro models that recapitulate the in vivo physiological epithelium has hampered the understanding of the single and coinfection processes that occur at the cervical epithelial barrier.

Our recently established human ectocervical organoid model that faithfully recapitulates the in vivo epithelial architecture of stratified squamous epithelium with its tissue-specific characteristics is a near-physiological in vitro model to investigate pathogenesis mechanisms[3,4]. Since HPV is the etiological agent of cervical cancer development and is often associated with simultaneous infections with *C. trachomatis*[10,36,37], the ectocervical organoids from HPV negative healthy donors are ideal tools to investigate the unexplored connection and the impact of HPV, *C. trachomatis* and coinfections on the host cells. The persistence of HPV triggered by integrating viral oncogenes E6 and E7 into the host genome predisposes host cells for malignant transformation[38–40]. Our study simulated the impact of HPV16 E6E7 by integrating these oncogenes into the human ectocervical epithelial stem cell genome. Notably, these human ectocervical stem cells containing HPV16 E6E7 developed the characteristics of CINs. The organoids derived from these stem cells show increased epithelial regeneration by enhanced proliferation and differentiation, nuclear atypia with varied nuclear size, and altered adherens junctions. However, they retain the capacity to self-renew, organize and differentiate to form mature squamous stratified organoids similar to healthy tissue with basal p63 + Ki67+ stem cells, p63 + Ki67-parabasal, and p63- Ki67- differentiated cells. Despite Chlamydia being one of the four most frequent sexually transmitted pathogens, its infection process in complex ectocervical stratified epithelial tissue is unknown. This study systematically illustrated the temporal and spatial infection process in the multilayered ectocervical organoids recapitulating the physiological infections. Chlamydia can infect ectocervical stem cells and complete its biphasic life cycle. *C. trachomatis* infection propagates in stem cells, spread to differentiated layers, and is ultimately released into the lumen. Our study provides insights into the influence of HPV E6E7 on *C. trachomatis* development. Interestingly, HPV E6E7 slows down the *C. trachomatis* developmental life cycle by inhibiting the redifferentiation of RBs into EBs and induces persistence.

Our global transcriptomic analysis revealed that HPV E6E7 and *C. trachomatis* elicit distinct host cell transcriptional programs promoting stemness and preneoplastic molecular phenotype. While TNF-mediated immune response is upregulated by both pathogens, *C. trachomatis* distinctively elicit IL-17 and NF-kappa B signaling inflammatory response. Chlamydia also markedly interferes with oxidative phosphorylation, RNA regulation, RNA processing, and upregulated MAPK pathways. Strikingly, superinfection of HPV16 E6E7 expressing cells with *C. trachomatis* suppresses a significant number of host genes induced by HPV E6E7, thus overwriting the host transcriptional program.

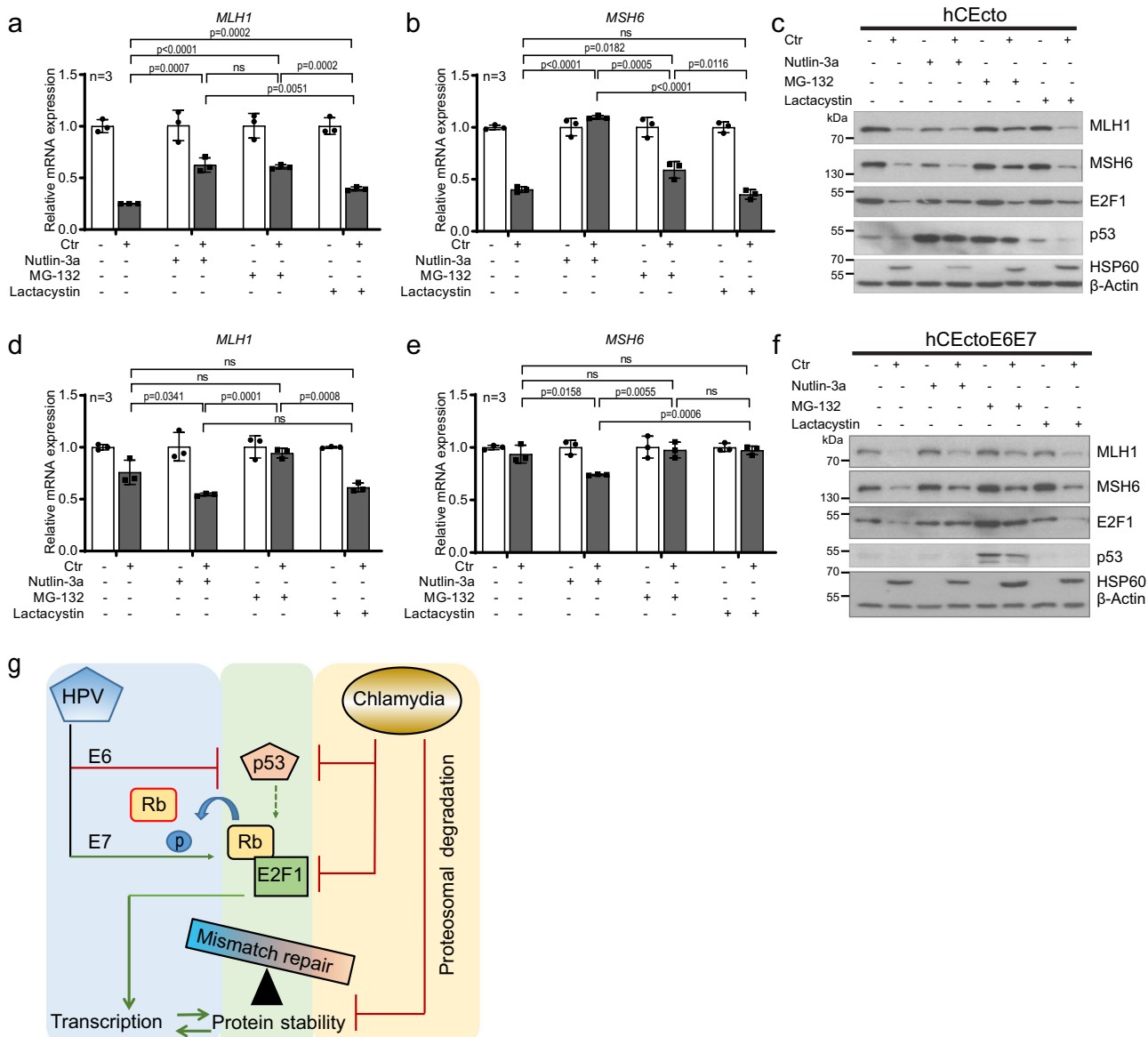

**Fig. 5 Chlamydia and HPV E6E7 coinfection regulates MMR distinctly at transcriptional and translational levels.** hCEcto and hCEcto E6E7 2D stem cells were infected with Ctr for 48 h with or without additional treatment with Nutlin-3a (10 μM) or Lactacystin (10 μM) from 2hpi or 24hpi with MG-132 (5 μM). **a**, **b** Shown is the relative mRNA expression of *MLH1* (**a**) and *MSH6* (**b**) analyzed by qRT-PCR from hCEcto cells. Data represent mean ± SD of three replicates normalized to uninfected controls from a representative of two independent experiments. Statistical significance was calculated using two-sided *t* test, *P*-values are indicated; ns, no significance. (**c**) Immunoblot analysis of lysates from hCEcto cells for indicated proteins and Chlamydia HSP60 and β-actin as a loading control. **d**, **e** Shown is the relative mRNA expression of *MLH1* (**d**) and *MSH6* (**e**) analyzed by qRT-PCR from hCEcto E6E7 cells. Data represent mean ± SD of three replicates normalized to uninfected controls from a representative of two independent experiments. Statistical significance was calculated using two-sided *t* test, *P*-values are indicated; ns, no significance. **f** Immunoblot analysis of lysates from hCEcto E6E7 cells for indicated proteins and Chlamydia HSP60 and β-actin as a loading control. Data from (**c**) and (**f**) represent two biological replicates. **g** Model depicting the mechanistic regulation of MMR by HPV E6E7 and Chlamydia infections. Source data are provided as a Source Data file.

We found that HPV E6E7 activates the E2F family of transcription factors, and their downstream-regulated genes involved in several cellular and genome surveillance pathways. In line with previous studies[41,42], HPV E6E7 was also found to increase the ATM-dependent homologous recombination pathway. Interestingly, Chlamydia superinfection of these cells suppresses E2F mediated regulation of DNA repair pathways and quiescence. Multiple signatures of mutational processes are identified in cervical cancer samples, with two signatures associated with defective MMR (https://cancer.sanger.ac.uk/cosmic/signatures_v2). Our data showed that in contrast to HPV E6E7, *C. trachomatis* inhibits the MMR pathway distinctly at the transcriptional and post-translational levels and further suppresses the HPV-induced MMR activities. We revealed that *C. trachomatis* inhibits MMR gene expression by proteasomal degradation of transcriptional factor E2F1. Suppressed MMR pathway coupled with the increased DNA damage induced by *C. trachomatis* could accelerate the mutagenic activity in the context of proliferation signals. The impact of HPV and Chlamydia's co-persistence in a stem cell could, thus, be deleterious to cellular and genomic stability and promote neoplastic progression.

Our cervical 3D organoids provide the much-needed near-physiological in vitro models to investigate various facets of cervix biology, including the influence of FRT infections and

coinfections and their molecular mechanisms. The long-term culturability of these organoids and the ability to genetically manipulate them opens avenues of approach to investigate the initiation, progression, and outcome of chronic infections in an authentic preclinical setting. By utilizing this powerful development, we show here the complex tripartite interactions of epithelial tissue, viral and bacterial coinfection, and the impact on host cell response and fate. Thus, this study goes beyond state-of-the-art and shows how multiple or sequential infections contribute to pathogenesis and drive cells towards cancer progression.

## Methods

**Patient material.** Human cervical samples were provided by the Department of Gynecology, Charité University Hospital, and Vivantes Auguste-Viktoria Klinikum, Berlin. Usage for scientific research was approved by the Ethics Commision of the Charité University Hospital, Berlin, Germany (EA1/059/15); informed consent to use their tissue for scientific usage was obtained from all donors. The study complies with all relevant ethical regulations regarding research involving human participants. Biopsies were sourced from standard surgical procedures for benign gynecological disease. Only anatomically normal tissue biopsies from anonymous female donors aged between 35 and 75 were processed within 2–3 h after removal.

**Human ectocervical primary cell isolation, cultivation, and maintenance.** Isolation of epithelial stem cells from the human cervix, cultivation, and maintenance in 2D and organoids was performed as described in[3,4]. In brief, tissue biopsies were washed, and the epithelial progenitors were isolated by mincing and enzymatic digestion using collagenase type II (0.5 mg/mL) (Calbiochem, #234155) (2.5 h, 37 °C). The dissociated cells were pelleted (5 min, 1000 $g$, 4 °C), resuspended in TrypLE (Gibco, #12604021), and incubated in a shaker (15 min, 37 °C). Primary cells were expanded in collagen-coated (Sigma, #C3867) tissue culture flasks until they reached 70–80% confluence in human ectocervical primary cell medium (Advanced DMEM/F-12 (Invitrogen, 12634) supplemented with 12 mM HEPES, 1% GlutaMax, 1% B27, 1% N2, 10 ng/mL human EGF (Invitrogen, 15630-056, 35050-038, 17504-044, 17502048, PHG0311), 0.5 μg/mL hydrocortisone (Sigma, H0888-1G), 100 ng/mL human noggin (Peprotech, 120-10 C), 100 ng/mL human FGF-10 (Peprotech, 100-26-25), 1.25 mM N-acetyl-L-cysteine (Sigma, A9165-5G), 2 μM TGF-β receptor kinase Inhibitor IV (SB431542, Calbiochem), 10 μM ROCK inhibitor (Y-27632) (Sigma, Y0503), 10 mM nicotinamide (Sigma, N0636), 10 μM forskolin (Sigma, F6886) before seeding ~20,000 cells/50 μL in Matrigel (BD, #356231) for culturing organoids or for maintenance of 2D stem cells on lethally irradiated J2-3T3 fibroblast feeder cells. For propagation, 2D stem cells were reseeded onto freshly irradiated J2-3T3 every 1 to 2 weeks at a 1:5 ratio. Ectocervical organoids were split every 2 weeks at a ratio of 1:5 by enzymatically splitting with TrypLE and mechanical fragmentation with a fire-polished glass Pasteur pipette by vigorous pipetting.

For 2D infection experiments of human ectocervical epithelial (hCEcto) stem cells, the cells were subjected to differential trypsinization to separate fibroblasts from epithelial cells, and only epithelial cells were seeded on collagen-coated cell culture plates. Plates were coated with collagen type I at diluted 1:100 ratio with PBS (solution from rat tail) (Sigma-Aldrich, #C3867) for 1 h at 37 °C. Cells were incubated at 37 °C for 24 h before infection.

**Cell lines.** HeLa cells (ATCC® CCL-2™), End1 E6E7 cells (ATCC® CRL-2615™), 293T cells (Invitrogen-10938-025), and J2-3T3 cells (mouse embryonic fibroblasts) (kind gift from Craig Meyers) were maintained in HEPES-buffered Dulbecco's modified Eagle's medium (DMEM) (Gibco, 10938-025) supplemented with 10% fetal calf serum (FCS) (Biochrome, S0115), 2 mM glutamine (Gibco, 25030081), and 1 mM sodium pyruvate (Sigma, S8636), at 37 °C in a humidified incubator containing 5% CO2. Human keratinocytes containing episomal HPV16 (KC-Epi)[43], a kind gift from Craig Meyers, Department of Microbiology and Immunology, Penn State University School of Medicine, Hershey, Pennsylvania 17033, USA. W12-20850 cells containing HPV16 in the episomal state (W12-Epi)[44] and W12-20831 cells where HPV16 is integrated (W12-Int)[45], a kind gift from Paul F. Lambert, University of Wisconsin-Madison, Madison, WI, USA, were cultured in E-medium supplemented with 3.07% sodium bicarbonate (Fluka, 71628), 180 μM adenine (Sigma, A-8626) and 1% nystatin (Sigma, N1638) in the presence of growth-arrested J2-3T3 cells or embedded in Matrigel.

**_C. trachomatis_ infections of human ectocervical stem cell.** _C. trachomatis_ L2 (ATCC, #VR-902B) infection experiments in 2D stem cells were performed at a multiplicity of infection (MOI) of 5 in ectocervical primary cell medium. The medium was exchanged at 2hpi, and cells were grown at 35 °C in 5% CO2 in a humidified incubator for 24 h pi to 48hpi and treated with chemical compounds depending on the experiment.

**_C. trachomatis_ infections of human ectocervical organoids.** For 3D ectocervical organoid _C. trachomatis_ infection experiments, human ectocervical organoids were grown for 5 days in a 24-well plate at 37 °C in 5% CO2 in human ectocervical primary cell medium without antibiotics. Then, the media was removed, and Matrigel was dissolved by adding 1 mL ice-cold ADF and pipetting up and down. The resuspended organoids were divided into two 15 mL tubes for infection and mock control. To release organoids completely from Matrigel, another 4 mL of ice-cold ADF was added to each tube. After centrifugation at 300 $g$ for 5 min at 4 °C, the supernatant was discarded, resuspend the pellet in 250 μL medium containing diluted Ctr stock to yield the approximate MOI of 10. Organoids were incubated for 2 h at 35 °C with 5% CO2 with gentle shaking at 100 rpm. After infection, organoids were centrifuged at 300 $g$ for 5 min, and the pellet was resuspended in Matrigel (50 μL/well) and kept at 35 °C with 5% CO2 in a humidified incubator. Once Matrigel was polymerized, a fresh human ectocervical primary cell medium was added, and the infection was allowed to proceed until the desired time.

**Infectivity assay.** hCEcto and hCEcto E6E7 stem cells grown in collagen-coated six-well plates were infected with _C. trachomatis_ L2 (MOI = 1) for 48 h and then scraped and transferred into a 15 mL tube containing sterile glass beads and lysed by vortexing for 4 min at full speed. To determine the infectivity from organoid infection experiments, Matrigel was first dissolved with ice-cold ADF, and the suspension was centrifuged (5 min at 300 $g$, 4 °C). Then the supernatant was discarded, and the pelleted organoids were resuspended in 1 mL of DMEM supplemented with 5% FCS (infectivity medium) and 1 mL of sterile glass beads followed by 4 min vortexing at full speed to lyse the cells. The lysates dilutions from 2D stem cells and organoids were then processed similarly by transferring onto HeLa cells seeded 1 day before in 96-well plates and incubated for 2 h. The lysates were discarded, an infectivity medium was added, and cells were incubated overnight at 35 °C with 5% CO2. Cells were fixed with ice-cold methanol 200 μL/well at 4 °C and immunostained with _Ctr_-major outer membrane protein (_Ctr_- MOMP KK12) specific antibody, and Cy3 labeled secondary antibody. Host cell nuclei were stained with Hoechst. The number of Chlamydial inclusions, inclusion size, and the number of host cells were analyzed with an automated microscope and ScanR Analysis Software v2.7.1 (Olympus Soft Imaging Solutions).

**Antibodies and Chemicals.** Antibodies and chemicals were obtained from the following sources: mouse-anti-p63 (4A4) (1:200, Abcam, #ab735), mouse-anti-E-Cadherin (1:100, BD Biosciences, #610181), rabbit-anti-Ki67 (SP6) (1:200, Abcam, #ab16667), rabbit-anti-Cytokeratin 5-Alexa488 (1:300, Abcam, #ab193894), rabbit-anti-Cytokeratin 8 (1:200, Abcam, #ab59400), rabbit-anti-Loricrin (1:50, Abcam, #ab85679), mouse-anti-phospho γH2AX (Ser139) (1:500, Millipore, #05636), rabbit-anti-MSH6 (EPR3945) (1:1000, Abcam, #ab92471), rabbit-anti-MLH1 (EPR3894) (1:10000, Abcam, #ab92312), mouse-anti-β-Actin (1:10000, Sigma, #014M4759), mouse-anti-E2F1 (8G9) (1:250, Abcam, #ab 135251), mouse-anti-Rb (4H1) (1:2000, Cell Signaling, #9309), rabbit-anti-pRb (Ser807/811) (1:1000, Cell Signaling, #9308), mouse-anti-p53 (DO-1) (1:500, Santa Cruz, #sc-126), mouse-anti-Chlamydia Hsp60 (A57-E4) (1:500, Enzo Life Sciences, #ALX-804-071-R100 and 1:1000 GeneTex, #GTX25486) and goat-anti-Chlamydia MOMP (1:500, AbD Serotec, #1990-0804), mouse monoclonal species-specific KK-12 IgG2a _Ctr_ (anti-MOMP) (1:10000, D. Grayston, University of Washington, Seattle, WA, USA). Secondary antibodies used for immunofluorescence were donkey-anti-goat Alexa Fluor 488 (Jackson ImmunoResearch, #705-545-147), donkey-anti-rabbit-Cy3 (Jackson ImmunoResearch, #711-166-152), donkey-anti-mouse Alexa Fluor 647 (Jackson ImmunoResearch, #715-605-150), goat anti-mouse-Cy3 (Dianova, #115-165-006), donkey-a-goat- Dylight 647 (Dianova, #705-605-003) (1:150) and donkey-anti-mouse Alexa Fluor 488 (Dianova, #715-454-151). Hoechst (1:2000, Sigma, #B2261) and Draq5 (1:1000, Thermo Scientific, #62252) were used to label DNA. For immunoblot analysis, secondary antibodies conjugated to horseradish peroxidase (HRP) sheep-anti-mouse IgG-HRP (Amersham Biosciencesa, #NA931) and donkey-anti-rabbit IgG-HRP (Amersham Biosciencesa, #NA934) were used at 1:2000 dilution. Following chemicals were obtained from Sigma-Aldrich: Proteasome inhibitor MG132 (#M7449), Proteasome and Chlamydial protease-like activity factor (CPAF) inhibitor Lactacystin (#L6785), and tumor suppressor p53 (TP53) stabilizer and MDM2 inhibitor Nutlin-3a (#N6287).

**RNA and DNA isolation.** RNA and DNA were isolated using the Allprep and RNeasy Mini Kit (Qiagen, #80204, #74104) according to the manufacturer's protocol.

**Polymerase chain reaction (PCR) and agarose gel electrophoresis.** The presence of genes encoding for HPV E6E7 was detected using the PCR. The PCR mixture contained 100 ng DNA template, 0.5 μL forward primer (10 μM), 0.5 μl reverse primer (10 μM), 0.5 μl dNTPs (10 mM) (Thermo Scientific, #R0182), 2.5 μL 10x NEB buffer (BioLabs, #B9004S), 1 μL NEB Taq polymerase (5000 U/mL) (BioLabs, #M0267S), 1 μL MgCl2 (25 mM) (Promega, #A351H) and the reaction was made up to 25 μL with H2O. The PCR cycling conditions were as follows: initial denature of 5 min at 95 °C followed by 35 cycles at 95 °C for 30 s, at 55 °C for 30 s and at 72 °C for 1 min. The final extension was at 72 °C for 5 min. PCR products were directly processed using agarose gel electrophoresis. DNA fragments

were separated in 1.5% agarose (Biozym, #840004) in 0.5% v/v TBE buffer supplemented with ethidium bromide solution at 120 V for 60 min. Up to 15 μL of PCR product was loaded into the agarose gel using a 6x DNA loading buffer (Fermentas, #R0611). Primers for HPV testing were described in[46]. Following primers for GAPDH were used as controls: Forward: 5′-GGTATCGTGGAAG-GACTCATGAC-3′, Reverse: 5′-ATGCCAGTGAGCTTCCCGTTCAG-3′; The primers were purchased from Sigma-Aldrich and diluted to 10 μM. See Supplementary Fig. 6 for full scan of gels.

**Quantitative real-time polymerase chain reaction (qRT-PCR) Analysis.** Relative RNA levels were determined by qRT-PCR using Power SYBR® Green RNA-to-CT™ 1-Step Kit (Thermo Fisher, #4389986), StepOnePlus ™ Real-Time PCR System (Applied Biosystems TM), and StepOneTM Software (v2.3, Applied Biosystems). The reaction mixture contained 25 ng RNA, 12.5 μL SYBR® green mix, 1.84 μL H2O, 0.16 μL Reverse Transcriptase enzyme mix and 0.5 μL Primer mix (10 μM) and was subjected to the following PCR cycler program: 30 min at 48 °C; 10 min at 95 °C; followed by 40 cycles of 15 s at 95 °C and 60 s at 60 °C. The relative expression levels of all genes were determined by normalizing to mRNA expression of housekeeping gene Valosin Containing Protein (VCP). All samples were measured as triplicates. Following primers were used: MSH6- Forward: 5′-CCCCAC-CAGTTGTGACTTCT-3′, Reverse: 5′-TGTTGGGCTGTCATCAAAAA-3′; MLH1- Forward: 5′-TGAGCAGGGACATGAGGTTCTC-3′, Reverse: 5′-ACTAAGCTTGGTGGTGTTGAG-3′; VCP- Forward: 5′-AGCATTGACC-CAGCTCTACG-3′, Reverse: 5′-TCTCATTGGCTACCTGTTCCAG-3′; E2F1- Forward: 5′-AGATGGTTATGGTGATCAAAGCC-3′, Reverse: 5′-ATCT-GAAAGTTCTCCGAAGAGTCC-3′; RB- Forward: 5′-CTCTCGTCAGGCTT-GAGTTTG-3′, Reverse: 5′-GACATCTCATCTAGGTCAACTGC-3′; TP53- Forward: 5′-CCTCCTCAGCATCTTATCCGA-3′, Reverse: 5′-TGGTACAGTCA-GAGCCAACCTC-3′. The primers were purchased from Sigma-Aldrich and diluted to 10 μM. The melting temperature was determined to be 60 °C for all primers.

**Sample preparation for Western blotting.** Total protein extracts from cells or organoids grown in six-well plates treated as per experimental requirements were prepared by incubating in 200-300 μL of SDS lysis buffer (3% 2-mercaptoethanol, 20% glycerin, 0.05% bromophenol blue (AppliChem, #A2331,0005), 3% SDS). The lysates were collected and boiled at 95 °C for 10 min. Samples were stored at −20 °C until required for Western blotting. See Supplementary Fig. 7 for full scan of blots.

**Generation of HPV16 E6E7 expressing hCEcto primary stem cells by lentiviral manipulation.** Replication-deficient lentiviral particles carrying HPV16 E6E7 or Luciferase gene were produced by transfection of 293 T cells with pLXSN HPV16 E6E7 (Addgene, #52394) or pSRE-Luciferase plasmid from ATCC (#MBA-120) together with packaging plasmids pMD2.G (Addgene, #12259), psPAX (Addgene, #212260) and Fugene6 (Promega, #E2691) diluted in Opti-MEM™ (Gibco, #31985088). For a 10 cm dish, 15.6 μL Fugene6 was mixed with 192.4 μL OPTI-MEM and added to 2.6 μg lentiviral target plasmid, 1.95 μg psPAX2 packaging vector, and 0.65 μg pMD.2 G (VSVG) envelope vector diluted in 52 μL OPTI-MEM. The DNA-Fugene Mix was incubated for 20–30 min at RT before adding dropwise to the 293 T cells. Cells were incubated with the mix for 12–15 h at 37 °C and 5% CO2 before the medium was replaced with fresh DMEM. Two days post-transfection, lentiviral particles in the medium were harvested, filtered (0.45 μm), and used for hCEcto cell transduction. For transduction, hCEcto stem cells were grown on collagen-coated six well-plate to 50% confluence before they were treated with virus solution and Polybrene (1 μg/mL) (Sigma-Aldrich, #H9268) overnight at 37 °C. When cells reached ~90% confluence, they were selected by 0.5 μg/mL Blasticidin (Gibco) treatment.

**Single-molecule RNA in situ hybridization (smRNA-ISH).** Single-molecule RNA in situ labeling of 10 μm thick paraffin-embedded organoid sections was performed using RNAscope 2.5 HD Red Reagent kit (Advanced Cell Diagnostics, 322350) as described previously[47] employing HPV16-E7 (#463461) and HPV16-L1 (#315601) (Advanced Cell Diagnostics) probes. Bright-field images obtained were processed with ImageJ v1.51f and Adobe Photoshop v23.1 and compiled with Adobe illustrator.

**Immunofluorescence histochemistry.** Organoids and human tissue were fixed with 3.7% paraformaldehyde (PFA, Sigma-Aldrich, #4441244), dehydrated, embedded in paraffin, sectioned, and stained for confocal imaging using Leica TCS SP8 (Leica Microsystems GmbH) as described in our detailed protocols[4,48]. Cells grown on collagen-coated coverslips in 2D were fixed with 3.7% paraformaldehyde for 30 min at RT. Cells were permeabilized and blocked with 2% Triton X-100, 0.05% Tween 20, and 1% BSA (Biomol, #01400.1) in PBS overnight at 4 °C. Primary antibodies were diluted in 1% BSA, 0.05% Tween 20 in PBS and incubated for 1 day at 4 °C followed by three washes in PSB-T (0.1% Tween 20 in PBS) and 1 h incubation with secondary antibodies diluted in 1% BSA, 0.05% Tween 20 in PBS along with Hoechst or Draq5. Before mounting with Mowiol, coverslips were washed three times with PBS-T and once with PBS. Images were acquired on a

Leica TCS SP8 confocal microscope and were processed with ImageJ v1.51 f and Adobe Photoshop v23.1.

**Transmission electron microscopy.** To analyze *C. trachomatis* development in human ectocervical epithelial cells, electron micrographs of infected and uninfected organoids and 2D stem cells were taken. Samples were prepared as described previously[49]. Briefly, at different time points after *C. trachomatis* infection, cells were fixed with 2.5% glutaraldehyde for 2 days at 4 °C. Cells were postfixed with 0.5% osmium tetroxide and tannic acid, contrasted with uranyl acetate, dehydrated in ascending ethanol series and infiltrated via styrene in epoxy resin. Samples were embedded in molds and allowed to polymerize at 70 °C. Electron micrographs were taken from 60 nm sections, contrasted with lead citrate, with a Leo 906E transmission electron microscope operated at 100 kV acceleration voltage (Zeiss, Oberkochen, DE) using a side-mounted digital camera (Morada, SIS-Olympus, Münster, DE).

**Mismatch DNA repair assay.** Fluorescent and nonfluorescent reporter plasmids used for measuring DNA repair capacity[29,50] were kindly provided by Dr. Zachary Nagel (Harvard T.H. Chan School of Public Health). ~70% confluent hCEcto and hCEcto E6E7 2D stem cells were washed with PBS, followed by TrypLE incubation 15 min at 37 °C to obtain single cells. For each sample, around 0.25 × 10⁶ cells were combined with a reporter plasmid mixture containing 0.5 ng of plasmid, that contains a G:G mismatch (G229C) (Table 1), in a total volume of 100 μL/ reaction and electroporated using an Amaxa ™ Basic Nucleofector ™ Kit for Primary Mammalian Epithelial cells (Lonza, #VPI-1005) and the Nucleofector™ 2b Device (Lonza) with the preset program T-20. Transfection efficiency in each experiment was measured by simultaneous transfection of fluorescent reporter plasmid mPlum. Following electroporation, 1 mL of human ectocervical primary cell medium was added, and the reaction mixture was divided into two tubes for infection and mock control. *C. trachomatis* infection was performed at a MOI 5. The inoculum was added after Nucleofection to a tube, followed by 1 h incubation at 37 °C with gentle shaking. The mock control was treated similarly. After infection, cells were pelleted by centrifugation (5 min at 1000 g; 4 °C), resuspended in human ectocervical primary cell medium, plated onto collagen-coated plates, and incubated for 24 h at 35 °C, 5% CO2 in a humidified incubator. Next, cells were washed with PBS, trypsinized with TrypLE for 15 min, and resuspended in 500 μL PBS containing Propidium iodide (PI, Thermo Scientific InvitrogenTM, #P3566) stain and transferred to 75-mm Falcon tubes with Cell Strainer Caps (Fisher Scientific). Cells were analyzed for fluorescence on a FACSymphony™ A5 (BDbiosciences) running the FACSDiva software v8.0.1 (BDbiosciences). Any cell debris, doublets, or aggregates were excluded based on side-scatter and forward-scatter properties. The gating strategy is provided in Supplementary Fig. 4e–j. The following fluorophores and their corresponding detectors were used: mOrange (505LP-580/20), mPlum (635LP-670/30), GFP (505LP-530/30). Compensation was set by using single-color controls. Data were quantified with FlowJo V10 (FlowJo, LLC). To calculate the relative reporter expression [% RE] the fluorescence signal of the total number of live cells appearing in the positive region of the fluorophores mOrange and mPlum were computed by multiplying the number of cells with the mean fluorescence intensity respectively. Then, to include the transfection efficiency, the fluorescence signal of mOrange was normalized to the fluorescence signal of mPlum for both uninfected and infected samples. The reporter expression shown is the normalized fluorescence signal of the infected sample relative to the uninfected sample. Note: In this assay the MMR reporter plasmid is non-fluorescent until the mismatch is repaired. Further the Chlamydia infected single-cell pool always also contains a fraction of uninfected cells. This fraction of uninfected cells might differ between experiments, and these uninfected cells within this fraction that are transfected with reporter plasmid would still be capable of performing MMR. Furthermore, the transfection efficiency might also vary from experiment to experiment. Due to these technical limitations in this assay, it is not possible to analyze only the reporter plasmid containing infected cells. Hence, the bar graphs of the infected samples in Fig. 4j, k show a big error bar. However, the results clearly demonstrate the decreasing trend of MMR in the cells with the activated reporter protein upon Chlamydia infection.

**Statistics.** Results are presented as the mean ± SD or min−max, with 25 and 75th percentile and median. For analysis, data sets were compared by two-sided *t* test or one-way ANOVA in GraphPad Prism v8 (GraphPad Software). Data are

**Table 1 Combinations of reporter plasmids and types of DNA damage used in each experiment.**

| Combination | mOrange | mPlum |
|---|---|---|
| #1 | Wild type | |
| #2 | | Wild type |
| #3 | Wild type | Wild type |
| #4 | G229C | Wild type |

considered statistically significant with a *p*-value ≤ 0.05. The exact *p*-values are, whenever possible, provided in the respective figure panel.

**Microarray expression profiling and data analysis**. 2D stem cells and organoids were pelleted and resuspended in 1 mL Trizol® reagent (Invitrogen™, #15596026). RNA was isolated according to the manufacturer's protocol, and the quantity of RNA was measured using a NanoDrop 1000 UV–Vis spectrophotometer (Kisker), and quality was assessed by Agilent 2100 Bioanalyzer with RNA Nano 6000 microfluidics kit (Agilent Technologies, #5067-1511). All Microarray experiments were performed as single-color hybridizations. Total RNA was amplified and labeled with the Low Input Quick-Amp Labeling Kit (Agilent Technologies, #5190-2305). In brief, mRNA was reverse transcribed and amplified using an oligo-dT-T7 promoter primer and labeled with cyanine 3-CTP. After precipitation, purification, and quantification, 0.75 µg labeled cRNA was fragmented and hybridized to custom whole-genome human 8 × 60K microarrays (Agilent-048908) according to the supplier's protocol (Agilent Technologies). Microarrays were scanned with 3 µm resolution (8 × 60K) using a G2565CA high-resolution laser microarray scanner (Agilent Technologies). Microarray image data were processed with the Image Analysis/Feature Extraction software G2567AA v. A.11.5.1.1 (Agilent Technologies) using default settings and the GE1_1105_Oct12 extraction protocol. The extracted single-color raw data files were background corrected, quantile normalized, and further analyzed for differential gene expression using Rstudio v. 1.4.1717 (Rstudio. Inc.). Microarray data from three independent infection experiments were combined. All genes differentially expressed with FDR < 5% (*p*-value ≤ 0.05) and log2 fold change (FC) < −1 or >1 for each comparison between any of four (hCEcto vs. hCEcto + Ctr; hCEcto vs. hCEcto + E6E7; hCEcto + E6E7 vs. hCEcto+E6E7 + Ctr, hCEcto + Ctr vs. hCEcto + E6E7 + Ctr) conditions were used.

For Venn diagram visualization with VENNY 2.1 [http://bioinfogp.cnb.csic.es/tools/venny/], statistically significant (*p*-value ≤ 0.05) up or down (FC > ± 2) regulated genes after *C. trachomatis* infection or HPV16 E6E7 expression in primary ectocervical cells were chosen.

**Analysis of KEGG pathways and Gene ontology**. For each condition, the infected cells are compared to the non-infected baseline (i.e., condition with no *Ctr* and no E6E7) or E6E7 transduced cells. For both 2D stem cells and organoids, we classify each item (gene, gene set, or regulon) regarding its result in *C. trachomatis* alone, coinfection, or E6E7 alone in a single class (e.g., Down-Down-Up). Corresponding tables were generated, and for genes, an enrichment was run for each class. All subclasses with less than or equal to 20 genes and all that have conflicting (Up-Down) results for probes within the same gene were excluded for further analysis.

**Gene Set Enrichment Analysis (GSEA)**. A pre-ranked analysis using the R package fGSEA was used that should give similar results to pre-ranked analysis in standard GSEA. The t-statistics from comparisons of ectocervical organoids (hCEcto vs. hCEcto + *Ctr*; hCEcto vs. hCEcto + E6E7; hCEcto + E6E7 vs. hCEcto+E6E7 + *Ctr* or hCEcto + *Ctr* vs. hCEcto + E6E7 + *Ctr*) were used to rank probes and enrichment of MSigDB V6.1 gene sets [http://software.broadinstitute.org/gsea/msigdb] (h.all.v6.1.symbols.gmt; c2.all.v6.1.symbols.gmt; c3.all.v6.1.symbols.gmt; c5.bp.v6.1.symbols.gmt; c6.all.v6.1.symbols.gmt; c7.all.v6.1.symbols.gmt) was computed using standard settings, collapsing probe sets within genes using the Max_probe method and using 5000 permutations. For further analysis, we kept only gene sets that were significant in at least one of the up or down regulated genes in the two comparisons mentioned above at FDR < 5%.

**Analysis of stemness and differentiation profile**. Stemness and differentiation-associated transcriptional changes in hCEcto and hCEcto E6E7 organoid either uninfected or infected with *C. trachomatis* were assessed by comparing to the DEG (*p*-value ≤ 0.05 and log2 FC<−1 or >1) between ectocervical stem cell and differentiated organoids[4].

**Profiling cervical intraepithelial neoplasia characteristics in coinfections**. To correlate gene expression patterns induced by HPV16 E6E7 and *C. trachomatis* coinfection with the gene expression profiles of CIN1-3, a microarray dataset of laser-captured epithelium from 100 cervical tissue specimens including normal (24), CIN1 (14), CIN2 (22), CIN3 (40)[20] was downloaded from the GEO with accession no. GSE63514 and analyzed using the R package maEndToEnd[51]. Background adjustment and quantile normalization were performed using the RMA algorithm provided by the package Oligo[52]. DEG with FDR < 5% (*p*-value ≤ 0.05) and log2 FC < − 1.5 or >1.5 from each of the two comparisons (hCEcto vs. hCEcto + E6E7; hCEcto + Ctr vs. hCEcto + E6E7 + Ctr) were utilized to check for their expression patterns in CIN samples.

**Reporting summary**. Further information on research design is available in the Nature Research Reporting Summary linked to this article.

## Data availability

Microarray data from this publication have been deposited in the National Center for Biotechnology Information Gene Expression Omnibus (GEO) under accession code GSE172426. Raw data associated with Figures can be found in the Supplementary Data respectively. The quantitative data of this study are available within the paper and its supplementary information files. Previously published microarray data that were re-analyzed here are available under accession codes GSE87076 and GSE63514. Uncropped images of the PCR gels and western blots labeled with identifying information are provided as Supplementary Figs. 6 and 7 respectively. All other data supporting the findings of this study are available from the corresponding author on reasonable request. Source data are provided with this paper.

## Code availability

Computational codes used for data analyses can be accessed at: https://github.com/ChumduriLab/Koster_Gurumurthy_et_al.Modelling_C.tr-_HPV_Co-infection

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

## Acknowledgements

The authors would like to thank late Jörg Angermann and Christiane Dimmler for technical help, Ina Wagner for the microarrays, and Diane Schad for help with graphics. The work was supported by the Max Planck Institute for Infection Biology and the University of Würzburg. C.C. acknowledges funding from DFG GRK2157. N.K. is supported by DFG DAAD and DFG GRK2157, P.G.P. is supported by DFG GRK2157, T.F.M. acknowledges funding from BMBF CINOCA, Z.N. acknowledges funding from U01ES029520 and P30ES00002. This publication was supported by the Open Access Publication Fund of the University of Wuerzburg.

## Author contributions

C.C. conceived and led the study. S.K., R.K.G., and C.C. designed the experiments. S.K., R.K.G., M.D., N.K, J.D, S.B, S.M.K, C.C., performed experimental work and analyzed the data. C.G., V.B. obtained electron micrographs, P.G.P, H.B., H.-J.M., S.K., R.K.G., and C.C., performed microarray analysis. T.F.M. provided the infrastructure and guidance. M.M. provided human samples. Z.N. provided the MMR reporter plasmids. S.K., R.K.G. and C.C. wrote the manuscript. All authors read the manuscript and provided feedback.

## Funding

## Competing interests

The authors declare no competing interest.
