## [Peer Review File · Nature Communications]

REVIEWER COMMENTS

Reviewer #1 (Remarks to the Author):

I think the authors did an excellent experimental work studying CT-HPV co-infection in ectocervix organoids.

I have only one comment:

In the results section, you described: "we established ectocervical organoids from healthy donors and tested for E6E7 genes from various hrHPV types. Shown are the ectocervical organoids from three donors found to be negative for E6E7 oncogenes of all tested hrHPV, including HPV 16 and 18 (Figure 1B)", but figure 1B only shows results for HPV 16 and 18, which high-risk genotypes you tested? Do you have results about other genotypes different from E6-E7- HPV16?

Reviewer #2 (Remarks to the Author):

Koster et al. describe a model of ectocervix organoids for analysing cross talk between HPV and chlamydia in co-infection and how they affect cellular activity and indicate that a weakening of genome maintenance mechanisms may be a result, thereby acting as a precursor to the development of malignant transformation. On the whole this is an extremely interesting and very well performed piece of work. There are however several issues which need to be addressed and several over simplifications which require clarification. Nonetheless this is a very important contribution.

1. Throughout the manuscript they use E6/E7 expressing cells and claim this as a model of HPV persistence. This is not correct. The model as presented is simply that of E6/E7 expression - in true viral persistence there would be episomal viral genomes and several other gene products. This needs correcting throughout the manuscript.
2. Changes in E6/E7 levels of expression have a major impact on disease progression. Some attempts should be made to quantify the level of E6/E7 expression in comparison to what would be seen in a normal CINI lesion or in cells which harbour episomal HPV and can complete a productive life cycle e.g W12 cells.

3. How much are the adverse effects of E6/E7 upon chlamydia related to changes in differentiation?
4. Figure 3. The levels of pRb seem to be unaffected in the E6/E7 cells when one would normally expect this to be degraded. Do the authors have an explanation for this?
5. Could this model system support a productive HPV life cycle as this is really fundamental. So the study would be strengthened greatly if episomal HPV sequences could be used if only to ascertain that viral DNA amplification could occur in the model. I realise future studies might also address effects of chlamydia in that context - and also upon viral episomal maintenance which is one of the major questions to emerge from this analysis.
6. Have they tried organotypic cultures containing HPV episomal genomes to assess chlamydia affects on viral life cycle?

Reviewer #3 (Remarks to the Author):

This manuscript addresses the role of the human papilloma virus (HPV) and the bacteria *Chlamydia trachomatis* in the transformation of ectocervical epithelium. This study builds upon previous work from the same lab using patient-derived ectocervical organoids; Here the authors describe their ability to manipulate ectocervical organoids by single or double infections to follow neoplastic transformation and transcriptional and post-translational changes. This study shows a distinct regulation of the DNA mismatch repair via MLH1 and MSH6 genes by HPV and *Chlamydia*. Novelty of HPV increasing the MMR repair pathway and activating the E2F family of transcription factor is moderate/low according to already previous work (Moody 2009 cited by authors, Teissier et al., *Oncogene* 2010).

Specific concerns:

1. The human ectocervical organoid model has been already published previously by this group and others (Lohmussaar et al., *Cell Stem Cell*) in 2021. As written, this is an important part of the publication, already published and therefore decreasing the novelty of this manuscript.
2. Organoid cultures should maintain long-term stem cell properties; therefore, it is unclear why authors culture their cells in parallel on 2D-3T3 feeders' cells. This culture probably changes the properties of the cells (even though microarray data show similarities) and it seems irrelevant to switch from 2D to 3D organoids.

3. Authors should define stem cells molecularly in their organoids culture, as well as stemness and finally long-term culture is vague as described.

4. Authors claim to reproduce the characteristics of CIN1 in ectocervical organoid expressing E6E7. This should be better characterized using in vivo transplantation assays and proper pathological analyses.

5. Many statements are made without quantifications:

- Immunostaining of CDH-1 in normal versus HPV-infected organoids should be quantified and not just mentioned as brightest

- Fig.1: The proliferation of the organoid infected with HPV should be quantified

- Fig 2K-N: Ki67 is mentioned as increased without quantification

- Fig 3J: The western blot claiming an increase E2F1 protein level upon E6E7 expression is not convincing

6. Loricrin staining in normal organoid is missing to compare with the HPV organoid

7. Page 7-8: statement of the way C. Trachomatis infection is progressing should be validated by data (this could be followed visually over time by live-cell imaging for example) "it progressed from the basal cells to daughter cells that self-renew by symmetric division andthe lumen"

8. The rationale of analyzing the 2D and organoid system at the transcriptional level is unclear

9. The impaired MMR repair efficiency (Figure 4J-K) shown by the decrease of the % of the reporter expression is not convincing.

Response to Reviewer Comments:

REVIEWER COMMENTS

Reviewer #1 (Remarks to the Author):

I think the authors did an excellent experimental work studying CT-HPV co-infection in ectocervix organoids.

Response: Thank you very much for your support and feedback.

I have only one comment:

In the results section, you described: "we established ectocervical organoids from healthy donors and tested for E6E7 genes from various hrHPV types. Shown are the ectocervical organoids from three donors found to be negative for E6E7 oncogenes of all tested hrHPV, including HPV 16 and 18 (Figure 1B)", but figure 1B only shows results for HPV 16 and 18, which high-risk genotypes you tested? Do you have results about other genotypes different from E6-E7- HPV16?

Response: We have tested for hrHPVs 31, 33, 35, 45, 51, 52, and 58 in addition to HPV16 and HPV18; as well as noncarcinogenic HPVs 6, 11, 40, 42, 43, 44, and 54 following the protocol described by Herfs et al. (PMID: 226899919). The results were negative for all tested HPV types in the donor's cells used for this study, as demonstrated by the lack of amplification in the PCR test; hence no bands were detected on the agarose gel. Therefore, the results were not included in the manuscript. However, we have included these data here for the reviewer's perusal. Additionally, we now state in the results section all the HPV types tested on Page 5; Line 111-116 and Page 19; Line 592-595.

If the reviewer suggests, we can also include it as supplementary data.

Reviewer #2 (Remarks to the Author):

Koster et al. describe a model of ectocervix organoids for analysing cross talk between HPV and chlamydia in co-infection and how they affect cellular activity and indicate that a weakening of genome maintenance mechanisms may be a result, thereby acting as a precursor to the development of malignant transformation. On the whole this is an extremely interesting and very well performed piece of work. There are however several issues which need to be addressed and several over simplifications which require clarification. Nonetheless this is a very important contribution.

Response: Thank you very much for the support and feedback. We have now added additional data and also modified the text to make it clearer to the readers.

1. Throughout the manuscript they use E6/E7 expressing cells and claim this as a model of HPV persistence. This is not correct. The model as presented is simply that of E6/E7 expression - in true viral persistence there would be episomal viral genomes and several other gene products. This needs correcting throughout the manuscript.

Response: We have now modified the text as suggested throughout the manuscript and explicitly mentioned E6E7 expressing cells instead of HPV persistence.

2. Changes in E6/E7 levels of expression have a major impact on disease progression. Some attempts should be made to quantify the level of E6/E7 expression in comparison to what would be seen in a normal CINI lesion or in cells which harbour episomal HPV and can complete a productive life cycle e.g W12 cells.

Response: We have performed the qRT-PCR analysis of HPV16 E6E7 negative, HPV16 E6E7 positive ectocervical cells in comparison to W12 cell lines with HPV genome in an episomal state (W12-epi) (PMID: 2467886) or with HPV E6E7

integration (W12-int) (PMID: 7707525) and primary keratinocytes containing episomal HPV16 genome (KC-epi) (PMID: 15110519). We found that E6E7 expression levels are higher in ectocervical and W12-int cells than W12-epi and KC-epi cells. Further, we found that the E6E7 expression in ectocervical organoids is relatively more than in the W12-int cells.

Now, this data is included in Figure 1k, and the explanation is included in the results section, Page 6; Lines 149-161.

3. How much are the adverse effects of E6/E7 upon chlamydia related to changes in differentiation?

Response: *We addressed the question in two different ways. First, to analyze the effect of HPV E6E7, Chlamydia, and coinfection on the stemness and differentiation in the ectocervical epithelium, we compared the cervical stem cell and differentiation transcriptional signatures that we previously defined in (Chumduri et al. 2021, PMID: 33462395) to the gene expression changes induced by E6E7 and Chlamydia infections. We found that E6E7 and Chlamydia infection both upregulate genes associated with stemness and down-regulate differentiation-associated genes. The new data is included as Figure 3i.*

In the second approach, we extracted the genes associated with skin development, epidermis development, keratinocyte differentiation, keratinization, epidermal cell differentiation, cornification that were oppositely regulated in GO terms in E6E7 and Chlamydia coinfections from supplementary Fig. 3h. The transcriptional regulation for these genes in HPV E6E7, Chlamydia, and coinfection scenario is shown in newly added Figure 3j as a heatmap. The data shows that a subset of the genes associated with these differentiation-associated GO terms are oppositely regulated while other genes are synergistically regulated.

Together the data shows that both HPV E6E7 and Chlamydia promote stemness and reduce differentiation. Based on these additional analyses, the text in the manuscript is modified on Page 10; Lines 284-295.

4. Figure 3. The levels of pRb seem to be unaffected in the E6/E7 cells when one would normally expect this to be degraded. Do the authors have an explanation for this?

Response:

- 1. The HPV16 E7 protein is predicted to bind to RB in co-immunoprecipitation studies (PMID: 2537532, PMID: 2556261). Further, it has been reported to promote RB degradation (PMID: 8840974; PMID: 11462030).*
- 2. However, we did not observe the degradation of RB at the protein level in our primary ectocervical cells expressing E6E7. Our observations are in agreement with other publications (PMID: 11803460 and http://archiv.ub.uni-heidelberg.de/volltextserver/8699/1/Komplette_Arbeit_18.06.2008.pdf). These studies showed that E7 overexpression alone does not induce degradation of pRB. Further, they demonstrated that E7 dependent RB degradation occurs only in the presence of histone deacetylase inhibitors SB and TSA.*
- 3. Nevertheless, one cannot rule out that the molecular profile (transcriptional activities) of the epithelial cell type might contribute to the difference in the RB regulation by HPV E7.*

We have now added this information in the results section on Page 9; Lines 264-269.

5. Could this model system support a productive HPV life cycle as this is really fundamental. So the study would be strengthened greatly if episomal HPV sequences could be used if only to ascertain that viral DNA amplification could occur in the model. I realize future studies might also address effects of chlamydia in

that context - and also upon viral episomal maintenance which is one of the major questions to emerge from this analysis.

Response: *We have now analyzed if the organoid model supports a productive HPV life cycle. For this, we generated organoids from the keratinocytes that harbor the HPV16 genome in an episomal state (KC-epi) and analyzed for expression of HPV early E7 and late L1 genes by smRNA-ISH. We also analyzed HPV E6E7 positive and negative ectocervical organoids as controls. We found the expression of E7 in all the cell types of both hCEcto E6E7 and KC-epi derived organoids. However, only KC-epi organoids showed L1 gene expression, essential for virion assembly, in the differentiated luminal cells indicating that organoids derived from cells harboring episomal HPV genome support a productive HPV life cycle. Thus, this data supports the feasibility of organoid models to further study native viral infections and coinfection studies.*

The newly generated data is included as Figure 1l-n. The corresponding explanation is included in the results section Page 6; Line 155-161.

6. Have they tried organotypic cultures containing HPV episomal genomes to assess chlamydia affects on viral life cycle?

Response: *In this study, we did not investigate the effect of Chlamydia infections on the HPV life cycle using organotypic/raft cultures.*

Reviewer #3 (Remarks to the Author):

This manuscript addresses the role of the human papilloma virus (HPV) and the bacteria *Chlamydia trachomatis* in the transformation of ectocervical epithelium. This study builds upon previous work from the same lab using patient-derived ectocervical organoids; Here the authors describe their ability to manipulate ectocervical organoids by single or double infections to follow neoplastic transformation and transcriptional and post-translational changes. This study shows a distinct regulation of the DNA mismatch repair via MLH1 and MSH6 genes by HPV and *Chlamydia*. Novelty of HPV increasing the MMR repair pathway and activating the E2F family of transcription factor is moderate/low according to already previous work (Moody 2009 cited by authors, Teissier et al., Oncogene 2010).

***Response:** Thank you very much for your support and feedback. We would like to clarify to the reviewer the novelty of this work.*

- 1. As mentioned in the manuscript, large-scale genomic studies that analyzed the mutational signatures and the mutational processes that contribute to these signatures in various cancers have revealed that two of the mutational signatures found in cervical cancer are indeed associated with defective mismatch repair (https://cancer.sanger.ac.uk/cosmic/signatures_v2). Further, some studies also show that cervical cancers have increased microsatellite instability, and this is due to defects in the MMR genes (PMID: 11586036). However, to date, no study has shown whether and how HPV infection modulates the MMR pathway.*
- 2. Moody 2009 et al. (PMID: 19798429) was cited by us to indicate the ability of HPV in increasing other DNA damage pathway such as ATM pathway. In that study, the authors specifically demonstrated the ability of HPV31 (using a cell line generated from a CIN lesion) in inducing the ATM pathway. This study did not address the MMR pathway regulation. However, the framing of the sentence in our manuscript might have been confusing. Hence,*

we have modified the sentence on Page 15; Lines 444-445. Thus, our study is the first to show how HPV and Chlamydia modulate this crucial DNA repair pathway during individual infections. Further, the novel aspect of our study is that coinfection with Chlamydia, in fact, suppresses HPV-induced genome maintenance pathways, including MMR, and that could potentially contribute to MMR mutational signatures observed in cervical cancers. Now we have modified the text in results section on Page 15; Line445-451.

3. In Teissier et al., Oncogene 2010 (PMID: 20639900), the E2F (1-6) transcription factor activation dynamics and the regulation of HPV18 E7 transcription in HPV18 positive cervical cancer HeLa cell line was studied.

However, they did not focus on the downstream target genes regulated by these transcription factors. Here again, the novelty of our study lies in the demonstration of the downstream genes of the transcription factors and the opposite regulation of E2F transcription factors in co-infections in primary cells.

Together, this work indeed is the first study to systematically investigate the effects of individual and co-infection of genital pathogens Chlamydia and HPV on the host cells in near-physiological epithelial cell models. Thus, providing plausible mechanistic principles underlying the cellular transformation and laying a foundation to study other co-infections of the female genital tract routinely observed in the clinic.

Specific concerns:

1. The human ectocervical organoid model has been already published previously by this group and others (Lohmussaar et al., Cell Stem Cell) in 2021. As written, this is an important part of the publication, already published and therefore decreasing the novelty of this manuscript.

Response: *This study is not addressing the development of the organoid model. As mentioned by the reviewer, the organoid development has already been published by us as a preprint (Chumduri et al. 2018, BioRxiv, <https://doi.org/10.1101/443770>) and recently in Nature Cell Biology, Chumduri et al. 2021 (PMID: 33462395). A similar study was also published recently by Lohmussaar et al., Cell Stem Cell in 2021 (PMID: 33852917). Accordingly, in the introduction, we mention the establishment of the organoid model and cite the papers on Page3; Lines 48-50. Also, in the results section, we have indicated that the organoid models have been established in the indicated studies and cited them on Page5; Line 100-102.*

We would like to highlight the following to the reviewers regarding this work's novelty that this is the first study investigating systematically the impact of HPV E6E7 and Chlamydia alone and coinfections in the near-physiological complex 3D ectocervical epithelial organoids derived from healthy human tissue.

In this study, for the first time, we have developed the protocols for genetic manipulation of these ectocervical primary cell-derived organoids. We have performed detailed investigations on the impact of HPV16 E6E7, Chlamydia alone, and coinfection on global transcriptional changes and signaling alterations. We identified how the co-infections might lead to the accumulation of mutations ultimately detected in the cancers by interfering with cellular DNA repair machinery in the stem cells of the ectocervix. We believe such analysis would not be possible using the immortalized or cancer cell lines. Thus, we strongly believe that the study indeed brings not only novel insights into the effects of co-infections on host cells; it also introduces the patient-derived healthy 3D organoid cultures of the cervix for such infection studies. This also opens up the possibility of studying the individual patients' (with different genetic makeup) responses to infection.

2. Organoid cultures should maintain long-term stem cell properties; therefore, it is unclear why authors culture their cells in parallel on 2D-3T3 feeders' cells. This culture probably changes the properties of the cells (even though microarray data show similarities) and it seems irrelevant to switch from 2D to 3D organoids.

Response: We would like to highlight to the reviewer that previously, we have demonstrated that in 2D cultures, we were able to enrich stem cells, and the longevity of these stem cells was maintained when cocultured in 3T3 feeders' cells with a unique cocktail of growth factors developed for ectocervical organoid culture. These 2D stem cells retain their capacity to form organoids. Further, the transcriptional profiles of 2D stem cells are similar to the stem cell compartment of the organoids (Chumduri et al. 2018 BioRxiv, <https://doi.org/10.1101/443770>; Chumduri et al., Nature cell Biology 2021(PMID: 33462395).

3D cultures maintained in Matrigel have both stem cells and differentiated cells. 2D stem cells would serve as a concentrated stem cell biobank and facilitate stem cell-specific studies, whereas 3D organoids recapitulate stem cell regeneration, differentiation processes and mimic cellular heterogeneity observed in the native tissue for studying infections and modeling diseases. Further, the 2D stem cell culture on the 3T3 cells offers a possibility to rapidly scale up the stem cell numbers, which can either be used to generate 3D organoids or directly used for infections studies. Since certain applications such as Mass spectrometry, Immunoprecipitation, immunopeptidomics, etc., require large quantities of cells, this could be achieved in a short-term using the 2D stem cells followed by organoid cultures in a cost-effective way.

However, it was imperative for us to demonstrate how the infections affect the 2D ectocervical stem cells compared to mature organoids. Hence, we have performed the analysis in both 2D stem cells and organoids. For instance, we show that the infections impact MMR predominantly in the stem cell compartment in organoids, which were also pronounced in 2D stem cell transcriptional signatures. Hence, these two approaches complement and synergize in elucidating the topological and cell-type response to infections.

We have now described this in the manuscript to make it clearer for the readers on Page 3; Lines 50-54.

3. Authors should define stem cells molecularly in their organoids culture, as well as stemness and finally long-term culture is vague as described.

Response: *Please see the response to points 1-2.*

The stemness and the long-term culturing for ectocervical epithelial-derived organoids have been defined in (Chumduri et al. 2018 BioRxiv, <https://doi.org/10.1101/443770>; Chumduri et al. Nature cell Biology 2021(PMID: 33462395).

4. Authors claim to reproduce the characteristics of CIN1 in ectocervical organoid expressing E6E7. This should be better characterized using in vivo transplantation assays and proper pathological analyses.

Response: *We would like to clarify that in our manuscript, we had stated that HPV16 E6E7 ectocervical cells show characteristics suggestive of CIN1 based on the features observed in organoids.*

As suggested by the reviewer, to investigate how the HPV E6E7 transduced cells behave in in vivo transplantation assays, one needs to systematically compare the HPV16 E6E7 organoids to human CINs-derived organoids as a positive control. For animal experiments and to derive organoids from CINs, animal permissions and human ethical permissions for precancer biopsies are essential, which is extremely difficult to obtain, even more so in these pandemic COVID times and is beyond the scope of this manuscript.

However, to address the reviewer's comment and to gain a deeper understanding of molecular characteristics of how HPV16 E6E7 induced effects correlate with CINs, we have performed a comparative analysis of HPV16 E6E7 and coinfection induced transcriptional alterations (up or downregulated genes) to the expression levels of

these genes in healthy and CIN1-3 tissues obtained from dataset published in (PMID: 26056290).

This analysis clearly demonstrates that the HPV E6E7 induces transcriptional changes that are similarly regulated in CINs. We observed that HPV E6E7 expression-induced changes not only correlate with that of CIN1 but also CIN2-CIN3. Therefore, we have now modified the sentence from "CIN1 to CINs" in the introduction, results, and discussion. Page 4; Line 83-85, Page 10; Line 294-299; Page 14; Line 419-420.

5. Many statements are made without quantifications:

- Immunostaining of CDH-1 in normal versus HPV-infected organoids should be quantified and not just mentioned as brightest

Response: *In the original version of the manuscript, we have described as "However, a significant alteration in the distribution of adherens junction marker E-cadherin (CDH1), responsible for maintaining cell polarity and differentiation, was observed in E6E7 positive organoids. E-cadherin expression in these organoids was brightest in the basal and parabasal layers, unlike in E6E7 negative organoids (Figure 1I)".*

However, to understand if E6E7 induces only redistribution of CDH1 or leads to changes in the overall protein level, we have now quantified the CDH1 protein levels by western blotting. The new data now included in Figure 1h revealed that CDH1 protein level is reduced in E6E7 expressing organoids. We have now described this observation in the results section on Page 6; Lines 138-140.

- Fig.1: The proliferation of the organoid infected with HPV should be quantified

Response: *We have now included the quantifications of the organoid proliferation ability by measuring the relative size of the organoids and organoid forming ability*

in HPV E6E7 expressing ectocervical organoids compared to E6E7 negative ectocervical organoids.

The newly generated data is included as Fig. 1i-j, and text in the results section is modified accordingly on Page 6; Lines 144-148.

- Fig 2K-N: Ki67 is mentioned as increased without quantification

Response: *The quantification of the Ki67 is now included in Figure 2s-t and in the results section on Page 8; Lines 222-227.*

- Fig 3J: The western blot claiming an increase E2F1 protein level upon E6E7 expression is not convincing

Response: *We have now quantified the E2F1 protein level, and the data revealing an increased expression of E2F is now included in Figure 3g.*

6. Loricrin staining in normal organoid is missing to compare with the HPV organoid

Response: *The images of Loricrin staining of normal organoids is now included in Supplementary Fig. 1b.*

7. Page 7-8: statement of the way C. Trachomatis infection is progressing should be validated by data (this could be followed visually over time by live-cell imaging for example) "it progressed from the basal cells to daughter cells that self-renew by symmetric division andthe lumen"

Response: *We have performed the live-cell imaging to demonstrate the progression of the Chlamydia infection in normal and HPV E6E7 expressing ectocervical organoids. The time-series data showing acquisition of inclusion in daughter cells in*

the basal layer as well as differentiated cell layers is now included as Figure 2a-b, and the text in the results section is updated accordingly on Page 7; Lines 171-183.

8. The rational of analyzing the 2D and organoid system at the transcriptional level is unclear

Response: *Please see the explanation in response to point 2.*

9. The impaired MMR repair efficiency (Figure 4J-K) shown by the decrease of the % of the reporter expression is not convincing.

Response: *The downregulation of the MMR pathway is predominantly seen in the cells infected with Chlamydia as shown in Fig. 4g-h. In this experiment, the restoration of the reporter protein expression (mOrange) upon repair of G:G mismatch is measured by FACS analysis. For this, we transfected hCEcto and hCEcto E6E7 cells using Lonza nucleofactor reagents with a G:G mismatch-containing nonfluorescent reporter plasmid and a control reporter plasmid (mPlum) simultaneously to estimate transfection efficiency. Transfection efficiency in our experiments is around 16%. Since this reporter plasmid is non-fluorescent until the mismatch is repaired, it is not possible to analyze only the reporter plasmid containing cells separately.*

Further, the Chlamydia infected single-cell pool always also contains a fraction of uninfected cells. This fraction of uninfected cells might differ between experiments, and these uninfected cells within this fraction that are transfected with reporter plasmid would still be capable of performing MMR. Furthermore, the transfection efficiency might also vary from experiment to experiment. Due to these technical limitations in this assay, it is not possible to analyze only the reporter plasmid containing infected cells. Hence, the bar graphs of the infected samples in Figures 4j-k show big error bar. However, the results clearly demonstrate the decreasing

trend of MMR in the cells with the activated reporter protein upon Chlamydia infection.

These results, in combination with the overall decrease in the MMR genes and proteins, demonstrated in the Chlamydia infected samples in Figure 4 c-f and specifically, in the inclusion containing organoids Figure 4g-h, it is clear that Chlamydia suppresses the MMR pathway.

REVIEWERS' COMMENTS

Reviewer #1 (Remarks to the Author):

The authors have answer my comments satisfactory.

It is not necessary include agarose gel images.

Reviewer #2 (Remarks to the Author):

The authors have done a very good job in answering my comments. I only have one residual query they may wish to comment on. In the organoid where L1 is detected it would seem that only very few cells are positive in and in one specific location. Could the authors speculate as to why this might be?

Reviewer #3 (Remarks to the Author):

1. The human ectocervical organoid model has been already published previously by this group and others (Lohmussaar et al., Cell Stem Cell) in 2021. As written, this is an important part of the publication, already published and therefore decreasing the novelty of this manuscript.

The response to this specific concern has not been addressed as the authors still claim in their revised manuscript line 79 that "This study addressed two key aspects: first we introduced a near-physiological human ectocervical organoid model to study the interaction of stratified epithelial tissue barrier and infection (.....)". This study only models dynamic of HPV16 and Chlamydia and their impact on the host cellular and molecular processes on a published system. This is up to the Editor to decide is the novelty is high enough to be published in Nat Comm. In that case the author should lower their tone in the writing.

The initial concern here was not that those authors have not cited their previous work and others, so the long response given has not addressed the first concern.

In their response to this first concern the author still write "Thus we strongly believe that the study indeed brings not only novel insights into the effects of co-infections on host cells" which here is acceptable; "it also introduces the patient-derived healthy 3D organoid cultures of the cervix for such infection studies" which again has been previously published

2. Organoid cultures should maintain long-term stem cell properties; therefore, it is unclear why authors culture their cells in parallel on 2D-3T3 feeders' cells. This culture probably changes the properties of the cells (even though microarray data show similarities) and it seems irrelevant to switch from 2D to 3D organoids.

OK concern addressed

3. Authors should define stem cells molecularly in their organoids culture, as well as stemness and finally long-term culture is vague as described.

OK concern addressed

4. Authors claim to reproduce the characteristics of CIN1 in ectocervical organoid expressing E6E7. This should be better characterized using in vivo transplantation assays and proper pathological analyses.

It is unclear why the authors do not show their new bioinformatic analysis that as they state in their response "clearly demonstrates that the HPV induces transcriptional changes that are similarly regulated in CINs"

5. Many statements are made without quantifications:

- Immunostaining of CDH-1 in normal versus HPV-infected organoids should be quantified and not just mentioned as brightest – OK concern addressed

- Fig.1: The proliferation of the organoid infected with HPV should be quantified- OK concern addressed

- Fig 2K-N: Ki67 is mentioned as increased without quantification - OK concern addressed

- Fig 3J: The western blot claiming an increase E2F1 protein level upon E6E7 expression is not convincing - OK concern addressed

6. Loricrin staining in normal organoid is missing to compare with the HPV organoid - OK concern addressed

7. Page 7-8: statement of the way C. Trachomatis infection is progressing should be validated by data (this could be followed visually over time by live-cell imaging for example) "it progressed from the basal cells to daughter cells that self-renew by symmetric division andthe lumen". OK concern addressed

8. The rationale of analyzing the 2D and organoid system at the transcriptional level is unclear. OK concern addressed

9. The impaired MMR repair efficiency (Figure 4J-K) shown by the decrease of the % of the reporter expression is not convincing.

The response of the authors describing the technical limitations of the assay should be integrated in the manuscript to clearly explain why the decrease presented in figure 4j-k is statistically not significant

REVIEWERS' COMMENTS

Reviewer #1 (Remarks to the Author):

The authors have answer my comments satisfactory.
It is not necessary include agarose gel images.

Response: Thank you very much.

Reviewer #2 (Remarks to the Author):

The authors have done a very good job in answering my comments. I only have one residual query they may wish to comment on. In the organoid where L1 is detected it would seem that only very few cells are positive in and in one specific location. Could the authors speculate as to why this might be?

Response: Thank you very much. HPV L1 is expressed in the late lifecycle stage and the differentiated cells. In the ectocervical organoids, the differentiation occurs towards the inner lumen. We found the L1 gene expression is also in a few cells that are probably in a conducive differentiation state.

Reviewer #3 (Remarks to the Author):

1. The human ectocervical organoid model has been already published previously by this group and others (Lohmussaar et al., Cell Stem Cell) in 2021. As written, this is an important part of the publication, already published and therefore decreasing the novelty of this manuscript.

The response to this specific concern has not been addressed as the authors still claim in their revised manuscript line 79 that "This study addressed two key aspects: first we introduced a near-physiological human ectocervical organoid model to study the interaction of stratified epithelial tissue barrier and infection (.....)". This study only models dynamic of HPV16 and Chlamydia and their impact on the host cellular and

molecular processes on a published system. This is up to the Editor to decide if the novelty is high enough to be published in Nat Comm. In that case the author should lower their tone in the writing.

The initial concern here was not that those authors have not cited their previous work and others, so the long response given has not addressed the first concern.

In their response to this first concern the author still write "Thus we strongly believe that the study indeed brings not only novel insights into the effects of co-infections on host cells" which here is acceptable; "it also introduces the patient-derived healthy 3D organoid cultures of the cervix for such infection studies" which again has been previously published

***Response:** Here, we would like to clarify that our study indeed introduces the patient-derived healthy 3D organoid cultures of the cervix for such infection studies" despite what has been published in the meantime. This study's in-depth mechanistic analysis of coinfections was published in the Biorxiv in April 2021. Around the same time, Lõhmussaar K et al. published that the cervix organoids can be infected with HSV, however, without an in-depth analysis of the infection outcomes.*

However, we have now modified the sentence on Page 4 as follows.

"This study addressed two key aspects; first, we introduced a near-physiological patient derived-ectocervical organoid model to study the interaction of stratified epithelial tissue barrier and individual and coinfection dynamics of HPV16 E6E7 and Chlamydia. Second, we systematically unraveled their impact on the host cellular and molecular processes".

2. Organoid cultures should maintain long-term stem cell properties; therefore, it is unclear why authors culture their cells in parallel on 2D-3T3 feeders' cells. This culture probably changes the properties of the cells (even though microarray data show similarities) and it seems irrelevant to switch from 2D to 3D organoids.

OK concern addressed

***Response:** Thank you.*

3. Authors should define stem cells molecularly in their organoids culture, as well as stemness and finally long-term culture is vague as described.

OK concern addressed

Response: Thank you.

4. Authors claim to reproduce the characteristics of CIN1 in ectocervical organoid expressing E6E7. This should be better characterized using in vivo transplantation assays and proper pathological analyses.

It is unclear why the authors do not show their new bioinformatic analysis that as they state in their response "clearly demonstrates that the HPV induces transcriptional changes that are similarly regulated in CINs"

Response: We want to clarify that the data (bioinformatic analysis) was indeed already included as Figure 3k in the previous revised submission. However, we missed to indicating the figure number in the last reviewer response.

5. Many statements are made without quantifications:

- Immunostaining of CDH-1 in normal versus HPV-infected organoids should be quantified and not just mentioned as brightest – OK concern addressed
- Fig.1: The proliferation of the organoid infected with HPV should be quantified- OK concern addressed
- Fig 2K-N: Ki67 is mentioned as increased without quantification - OK concern addressed

Response: Thank you.

- Fig 3J: The western blot claiming an increase E2F1 protein level upon E6E7 expression is not convincing - OK concern addressed

Response: Thank you.

6. Loricrin staining in normal organoid is missing to compare with the HPV organoid - OK concern addressed

Response: Thank you.

7. Page 7-8: statement of the way C. Trachomatis infection is progressing should be validated by data (this could be followed visually over time by live-cell imaging for example) "it progressed from the basal cells to daughter cells that self-renew by symmetric division andthe lumen". OK concern addressed

Response: Thank you.

8. The rationale of analyzing the 2D and organoid system at the transcriptional level is unclear. OK concern addressed

Response: Thank you.

9. The impaired MMR repair efficiency (Figure 4J-K) shown by the decrease of the % of the reporter expression is not convincing.

The response of the authors describing the technical limitations of the assay should be integrated in the manuscript to clearly explain why the decrease presented in figure 4j-k is statistically not significant

Response: We have now describe the following in the Methods section on page 23.

"Note: In this assay the MMR reporter plasmid is non-fluorescent until the mismatch is repaired. Further the Chlamydia infected single-cell pool always also contains a fraction of uninfected cells. This fraction of uninfected cells might differ between experiments, and these uninfected cells within this fraction that are transfected with reporter plasmid would still be capable of performing MMR. Furthermore, the transfection efficiency might also vary from experiment to experiment. Due to these technical limitations in this assay, it is not possible to analyze only the reporter plasmid containing infected cells. Hence, the bar graphs of the infected samples in Figures 4j-k show big error bar. However, the results clearly demonstrate the decreasing trend of MMR in the cells with the activated reporter protein upon Chlamydia infection".